

# Climatological study of a new air stagnation index (ASI) for China and its relationship with air pollution

Qianqian Huang, Xuhui Cai, Jian Wang, Yu Song, Tong Zhu

College of Environmental Sciences and Engineering, State Key Joint Laboratory of Environmental Simulation and Pollution Control, Peking University, Beijing, 100871, China

*Correspondence to:* Xuhui Cai (xhcai@pku.edu.cn)

**Abstract.** Air stagnation index (ASI) is a vital meteorological measure of the atmosphere's ability to dilute air pollutants. The original metric adopted by US National Climatic Data Center (NCDC) is found not well suitable to China, because the decouple between upper and lower atmospheric layer results in a weak link between the near surface air pollution and upper-air wind speed. Therefore, a new threshold for ASI is proposed, consisting of daily maximal ventilation in the atmospheric boundary layer, precipitation and real latent instability. In the present study, the climatological features of this newly defined ASI is investigated. It shows that the spatial distribution of the new ASI is similar to the original one; that is, annual mean stagnations occur most often in the northwest and southwest basins, i.e., Xinjiang and Sichuan basins (more than 180 days), and the least over plateaus, i.e., Qinghai–Tibet and Yunnan plateaus (less than 40 days). However, the seasonal cycle of the new ASI is changed. Stagnation days under new metric are observed to be maximal in winter and minimal in summer, which is positively correlated with air pollution index (API) during 2000–2012. The correlation between ASI and concentration of fine particulate matter (PM2.5) during January 2013 of Beijing is also investigated. It shows that the new ASI matches the day-by-day variation of PM2.5 concentration very well and is able to catch the haze episodes in that month.

## 1. Introduction

Air pollution has attracted considerable national and local attention and become one of the top concerns in China (e.g., Chan et al., 2008; Guo et al., 2014; Huang et al., 2014; Peng et al., 2016). It is also a worldwide problem, shared by the United States, Europe, and India, etc (e.g., Lelieveld et al., 2015; van Donkelaar et al., 2010; Lu et al., 2011). The initiation and persistence of air pollution episodes involve complex processes, including direct emissions of air pollutants, and secondary formation by atmospheric chemical reactions, etc. Meteorological background is also significant for its ability to affect the accumulation and removal or dispersion of air pollutants (e.g., Tai et al., 2010; Chen and Wang, 2015; Dawson et al., 2014). A substantial number of studies have been carried out on the abnormal meteorological conditions during haze events. They suggested that a persistent or slow


moving anti-cyclone synoptic system inclines to forms a haze episode (Ye et al., 2016; Leibensperger et al., 2008; Liu et al., 2013). Besides the driving force from synoptic scale, some local meteorological conditions are also favorable for heavy air pollutions. Low atmospheric boundary layer (ABL) height can act as a lid to confine the vertical mixing of air pollutants (Zhang et al., 2010; Ji et al., 2012; Bressi et al., 2013). Light winds lack the ability

to disperse air pollutants or transport them far away (Fu et al., 2014; Rigby and Toumi, 2008; Tai et al., 2010), and those pollutants blown by winds could directly contaminate the downwind zone (Wehner and Wiedensohler, 2003; Elminir, 2005). In addition, high relative humidity usually results in low visibilities through aerosol hygroscopic growth (Chen et al., 2012; Quan et al., 2011).

Air stagnation index (ASI), consisting of upper- and lower-air winds and precipitation, is designed to describe the ability of transport and dispersion of the atmosphere. A given day is considered stagnant when the 10 m wind speed is less than 3.2 m/s (i.e., near surface wind is insufficient to dilute air pollutants), 500 hPa wind speed less than 13 m/s (i.e., lingering anti-cyclones, indicating weak vertical mixing), and no precipitation occurs (i.e., no rain to wash out the pollutants) (Korshover, 1967; Korshover and Angell, 1982; Wang and Angell, 1999; Leung and Gustafson,

2005; Horton et al., 2014). Perturbation studies have proven that air stagnation positively correlates with ozone and particulate matter concentrations (e.g., Liao et al., 2006; Jacob and Winner, 2009). Hence, the US National Climatic Data Center (NCDC) monitors monthly air stagnation days for the United States since 1973 to indicate the temporal buildup of ozone in the lower atmosphere (http://www.ncdc.noaa.gov/societal-impacts/air-stagnation/).

Huang et al. (2017) followed the NCDC's metric of air stagnation and investigated the climatological mean features of it for China. It was found that the spatial distribution of annual mean stagnations is in good agreement with that of air pollution index (API). Basins in Xinjiang and Sichuan provinces are reported to experience heavy air pollutions most often (Mamtimin and Meixner, 2007; Wang et al., 2011; Chen and Xie, 2012; Li et al., 2015), and according to Huang et al. (2017), these two basins were identified as regions with most frequent stagnation

occurrence (50% days per year). However, the seasonal variation pattern of air stagnation barely correlated with that of API. Stagnations happen the most frequently in summer and the least in winter, whereas the API has the opposite seasonal pattern (Li et al., 2014). Why does the NCDC criteria fail to represent it? As the mid-tropospheric wind is the main driving force of air stagnations (Huang et al. 2017), we analyzed the 500 hPa wind speed in the most polluted days of Beijing when their API ranks in the 90[th] percentile or higher in the autumn and winter seasons

of each year during 2000–2012 (Fig. 1). It is found that in 75% of the most polluted days, the 500 hPa wind speed is stronger than 13 m/s. It suggests that air pollutions do not necessarily link with weak upper-air wind speed, and



thus weak 500 hPa wind speed is not an appropriate indicator for pollution occurrence.

In order to address this deficiency, Wang et al. (2017) tried to modify ASI by taking 10 m wind speed, ABL height and the occurrence of precipitation into consideration. Their air stagnation threshold is determined by a fitting equation which relates to $PM_{2.5}$ concentration, 10 m wind speed and ABL height. This equation varies with locations and changes over time. In this study, we propose a new air stagnation identification. Instead of relating our ASI directly to air pollution monitoring data, we put the thresholds on meteorological basis only and hope that the proposed ASI is a more universal one. The new ASI metric retains no-precipitation threshold to exclude wet deposition, gives up upper-air wind speed, replaces 10 m wind speed with ventilation in the ABL to represent both the horizontal dilution and vertical mixing scale of the atmosphere, and takes real latent instability of the surface atmosphere into consideration. The components of new ASI are introduced one by one in the following except precipitation threshold.

The depth of ABL is of much concern in air pollution studies, since it acts as the vertical mixing confinement of air pollutants, and in turn, determines surface concentrations of them. The ABL depth normally shows a diurnal cycle: a minimum at night and maximum in the afternoon (Stull, 1988). So is the vertical mixing extent of air pollutants. On the other hand, the strength of horizontal dilution (i.e., transport wind speed) is also significant in atmospheric dispersion and presents a similar diurnal pattern as ABL height. For the consideration of environmental capacity, mixing layer depth (MD) and transport wind in the afternoon are of great importance since they together represent the best dilution condition during the entire day (Holzworth, 1964; Miller, 1967). Therefore, ventilation in the maximum MD (referred to as ventilation hereinafter) is adopted in the newly defined ASI as a mixture to simultaneously indicate the largest extent of horizontal dilution and vertical mixing.

MD is not a quantity of routine meteorological observation (Liu and Liang, 2010). It is usually derived from profiles of the temperature and wind speed/direction as well as humidity from local and short-term observation project or experiment (Seibert et al., 2000). The methods used to detect or estimated MD includes: observation of sodar (Beyrich, 1997; Lokoshchenko, 2002), lidar (Tucker et al., 2009), ceilometer (Eresmaa et al., 2006; van der Kamp and Mckendry, 2010), wind profiler (Bianco and Wilczak, 2002), etc. Holzworth (1964, 1967) pioneers the work to obtain long term, large area maximum MD (MMD) information by "parcel method", in which MMD is interpreted as the height above ground of dry adiabatic intersection of the maximum surface temperature during the daytime with the vertical temperature profile from morning (1200 UTC for China) radiosonde observation.





Seidel et al. (2010) compared MMD results from parcel method along with six other methods using long-term radiosonde observations, and concluded that climatological MMD based on parcel method is warranted for air quality studies. Their further work (Seidel et al. 2012) compared the results derived from sounding data with other climate models, and provided a comparison of the MD characteristics between North America and Europe. These

two studies lay the groundwork for the global climatology of MD. Since then, the climatological characteristics of MD over regional (e.g., for Swiss plateau by Coen et al. (2014), for Indian sub-continent by Patil et al. (2013)) or global scales have been investigated (Von Engeln and Teixeira, 2013; Ao et al., 2012; Xie et al., 2011; Jordan et al., 2010).

MMD derived from parcel method is recommended for air pollution potential evaluation. However, in the presence of clouds, it agrees well with the depth of cloud base (Seidel et al., 2010), which is lower than the actual magnitude of vertical mixing extent of air pollutants. In order to compensate for this deficiency, we take the thermo-dynamical parameter convective available potential energy (CAPE) into consideration, implying the energy of ascension from the base to top of the cloud (Riemann-Campe et al., 2011). CAPE denotes the potential energy available to form

cumulus convection and thus serves as an instability index (Blanchard, 1998). It is a measurement of the maximum kinetic energy per unit air mass achieves through rising freely from the level of free convection (LFC) and the level of neutral buoyancy (LNB) (i.e., bottom and top of the cloud), since the virtual temperature of the buoyant parcel is higher than that of its environment (Markowski and Richardson, 2011). Holton (1992) indicated that when buoyancy is the only force of an air parcel, CAPE corresponds to the upper limit for the vertical component of the

velocity, as $CAPE = w_{max}^2/2$. However, high values of CAPE do not necessarily lead to strong convection (Williams and Renno, 1993). Its opposing parameter convective inhibition (CIN) describes the energy needed by the rising air parcel to overcome the stable layer between the surface and LFC (Markowski and Richardson, 2011; Riemann-Campe et al., 2009). CAPE stored in the atmosphere is activated if the value of CAPE is larger than that of CIN, that is, under real latent instability conditions. Therefore, climatological study of CAPE and CIN provides

an insight into the strength of atmospheric convection (Monkam, 2002).

In sum, unfavorable meteorological variables are critical to the occurrence of air pollution episodes. ASI from NCDC is designed to evaluate the air dilution capability, but is found to be inappropriate for China. So we propose a new ASI instead. Our aim in the present work is to investigate the climatological features of the newly defined

ASI, as well as its components, and examine the practicability of the new ASI by studying its relationship with air quality monitoring data, i.e., API data during 2000–2012, and concentrations of fine particulate matter ($PM_{2.5}$) in





January 2013 of Beijing.

**2. Data and Methodology**

**2.1 Data**

**2.1.1 Meteorological data**

Radiosonde data are used to calculate the daily ventilation. Soundings from all the international exchange stations
across China (95 stations) for the 30-year period (1985–2014) were obtained from the archive of the University of
Wyoming (available at http://weather.uwyo.edu/upperair/sounding.html). This dataset provides twice daily (0000
and 1200 UTC) atmospheric soundings, including the observed temperature, geopotential height, and wind speed

data at pressure levels, as well as additional derived variables CAPE and CIN.

Surface observational data are used to identify daily maxima of surface air temperature. We extracted hourly
observations of surface air temperature during 1985 to 2014 from Integrated Surface Database (ISD), provided by
NCDC. Data are accessible from FTP (ftp://ftp.ncdc.noaa.gov/pub/data/noaa). The entire ISD archive has been

previously processed through quality control procedures by NCDC, including algorithms checking for extreme
values and limits, consistency between parameters, and continuity between observations. More detailed information
regarding the quality control process can be found at https://www.ncdc.noaa.gov/isd.

Long-term (1985–2014) daily precipitation data were collected from China Meteorological Administration (CMA).

These           data           are           available           at
http://data.cma.cn/data/detail/dataCode/SURF_CLI_CHN_MUL_DAY_CES_V3.0.html.

For any investigation, we limited the study to the stations where the all quantities—radiosondes, surface
temperature observations and daily precipitation data—are available together. Therefore, air stagnations of 66

stations are analyzed in this study. Figure 2 shows that these 66 stations are fairly spread all over the contiguous
China, except that the Qinghai–Tibet Plateau particularly the western Tibet is not well sampled.

We conducted a general survey of the availability of valid data (radiosonde data, daily maximum temperature, and
daily precipitation are valid at the same time) for each of the 66 stations (Appendix A). It shows that among these



66 stations, datasets of 62 stations are available from January 1985 to December 2014, while datasets of the other 4 stations (Wenjiang, Jinghe, Chongqing, Shanghai) cover less than 30 years. The shortest duration is 9 years at Wenjiang station. The percentage of valid data are more than 96% of each station. In conclusion, the datasets are sufficient for us to conduct a climatological study of the new ASI on a countrywide basis.

### 2.1.2 Air quality data

To discuss the practical applicability of new ASI, daily API data (available at http://datacenter.mep.gov.cn) of four representative stations (Harbin, Urumqi, Beijing and Chongqing) during 2000–2012 are used. In order to exclude the influences of emission variations and chemical reactions to the greatest extent, and focus on the effects of meteorological conditions, we only analyse API data on those days when the primary pollutant is $PM_{2.5}$. The concentration of $PM_{2.5}$ is also used as another good metric for air pollution. We collected hourly $PM_{2.5}$ data of Beijing in January 2013 from the US Embassy (http://www.stateair.net/web/historical/1/1.html), and conducted a case study to examine the ability of new ASI to track day-by-day variation of air pollution.

### 2.2 Method

### 2.2.1 Daily MMD and ventilation

The parcel method (Holzworth, 1964) are used to calculate daily MMD. There are two steps in this calculation: 1) identify the daily maximum surface air temperature from hourly temperature data during 0800 to 2000 Beijing standard time (BJT; UTC+8); 2) starting from the maximal surface temperature, a dry-adiabatic line extends to intersect the temperature profile of morning sounding. Then the intersection height above ground level is MMD. Since the vertical resolution is coarse ($10^1$–$10^2$ m), temperature profiles from radiosonde are linearly interpolated to 1-m vertical intervals. It is noticed that this method may fail to determine the MMD sometimes when the temperature profile itself is close to a dry-adiabatic line (Lokoshchenko, 2002), or the height corresponding to the first valid value of the temperature is too high. These cases are treated as missing data and ignored.

Wind profile from 1200 UTC (i.e., 2000 BJT) sounding data is also interpolated to 1-m vertical grids, and ventilation is derived by integrating the wind speeds from surface to the top of the maximum mixing layer:

$$vent = \int_0^{MMD} u(z)dz \,, \tag{1}$$



where $vent$ is short for ventilation, z is the elevation above surface, $u(z)$ is the wind speed at z m above the ground. Ventilation is a direct measure of the dilution capacity of the atmosphere within the ABL. Higher values of ventilation indicate effective dilution.

From Equation (1), the average value of the wind speed through the MMD, i.e., the transport wind speed ($U_T$) writes:

$$U_T = {vent}/{MMD}.$$    (2)

### 2.2.2 CAPE and CIN

The values of CAPE and CIN are taken directly from the sounding files provided by University of Wyoming. Based on pseudo-adiabatic assumption, the formula for CAPE calculation (Murugavel et al., 2012) is:

$$CAPE = g \sum_{LFC}^{LNB} \frac{\Delta z(T_{vp} - T_{ve})}{T_{ve}},$$    (3)

where $T_{vp}$ designates the virtual temperature of lifted air parcel of the lowest 500 m of the atmosphere, $T_{ve}$ is virtual temperature of the environment, $\Delta z$ is incremental depth and g is gravitation constant. Only values of
CAPE calculated from 1200 UTC (i.e., 2000 BJT) profiles are considered for this analysis, representing the maximum convective potential energy of the day.

The value of CAPE is sensitive to the calculation method, such as the treatment of the freezing process (Williams and Renno, 1993; Thompkins and Craig, 1998; Frueh and Wirth, 2007). However, climatological mean features of
CAPE are not sensitive to the calculation method if one particular method is applied throughout (Myoung and Nielsen-Gammon, 2010; Murugavel et al., 2012). Therefore, our results are assumed to be insensitive to the method of CAPE calculation.

CIN is calculated through:

$$CIN = -g \sum_{Zsurface}^{LFC} \frac{\Delta z(T_{vp} - T_{ve})}{T_{ve}}.$$    (4)

It should be noted that values of CIN in the following discussion all refer to the absolute values of CIN.



### 2.2.3 New criteria for ASI

The new ASI is designed to better describe the atmospheric dispersion capability in ABL. A given day is considered stagnant when the value of daily maximum ventilation is less than 6000 $m^2$/s (Holzworth, 1972), the value of CAPE is less than that of CIN, and daily total precipitation is less than 1 mm (i.e., a dry day). The definition of air stagnation case is retained: 4 or more consecutive air stagnant days are considered as one air stagnation case (Wang and Angell, 1999). In order to show the spatial distribution of stagnation days and cases over continental China, results of 66 stations have been interpolated in space on the $2° \times 2°$ grid by cubic splines.

### 3. Climatology of new ASI components: ventilation, precipitation, and real latent instability

### 3.1 Ventilation climatology

The daily maximal ventilation condition is distributed with substantial regional heterogeneity (Fig. 3a). It is shown that ventilation is largest (more than 15000 $m^2$/s) over Qinghai–Tibet Plateau, which is the world's highest plateau with an average elevation exceeding 4500 m. The areas neighboring to the plateau, Yunnan Plateau and Inner Mongolian Plateau, also experience relative large ventilation (10000–15000 $m^2$/s). However, the south of China, where the population and industry are dense, exhibits the most unfavorable atmospheric ventilation conditions, especially the Sichuan Basin where the ventilation is lower than 4000 $m^2$/s.

Ventilation is a combined response of both the MMD and transport wind speed. It is noticed that regions with high ventilation also experience relatively high MMD and strong transport wind, whereas regions with poor ventilation conditions also exhibit low MMD and weak transport wind (Fig. 3). First of all, large MMD (more than 2000 m) is observed over Qinghai–Tibet–Yunnan Plateau, whereas low MMD (about 1000 m or less) in the southeast China (Fig. 3b). This distribution pattern is interpreted as the effect of terrain. High terrain elevation corresponds to thinner air and lower value of aerosol optical thickness, less precipitable water vapor amount, and stronger solar radiation (Chen et al., 2014; Zhu et al., 2010), which enhance thermal convections and lead to high MMD. These results agree well with the study of Liu et al. (2015) for Qinghai–Tibet–Yunnan Plateau. The spatial distribution of transport wind speed is displayed in Fig. 3c. It is noted that under the combined influence of westerly jet and topography (Schiemann et al., 2009), the transport wind is strong (more than 6 m/s) over plateaus and northeast plains, and weak (less than 4 m/s) over Xinjiang and Sichuan basins.



Four representative stations (Harbin, Urumqi, Beijing and Chongqing; shown in Fig. 2 as crosses) are selected to analyze the seasonal variation of ventilation, as well as the corresponding MMD and transport wind speed (Fig. 4). It is shown that patterns of the monthly variation of ventilation are different at different stations. Ventilations of Harbin and Beijing reach a peak in April, followed by a rapid fall to a low value in August. Subsequently, these

values increase to a secondary peak in October and then decrease to the minima in January. Correspondingly, MMD and transport wind speed also show a bimodal distribution pattern with two peaks: one in spring and another in autumn. For the north China, solar radiation is strong in summer; but significant rains and cloudy skies restrict the thermal convective processes and keep the MMD to a low value in monsoon season. In addition, the wind speed is typically weaker in summer and stronger in winter (Frederick et al., 2012). However, the winter solar radiation is

weak and constrains the development of atmospheric boundary layer. Consequently, weak transport wind and poor ventilation are observed in both summer and winter. The seasonal behavior of ventilation conditions of Urumqi is different. The monthly mean value of ventilation presents a unimodal distribution with the mean value ranging from 788 $m^2/s$ (January) to 15544 $m^2/s$ (June) through the year. The corresponding MMD and transport wind speed also show similar seasonal behavior, ranging from 334 m and 2.2 m/s in January to 2406 m and 6.2 m/s in June,

respectively. This may be attributed to the unique local climate of northern Xinjiang, where the climate is considered as semi-arid steppe (Peel et al., 2007). In Urumqi, short summer and long winter are the dominate seasons while spring and autumn are only transitional ones (Domroes and Peng, 1988). Monthly precipitation averaged over the recent 30 years (1985–2014) ranged between 12.2 mm (January) and 39.9 mm (May), demonstrating the arid climate of Urumqi (shown later in Fig. 6). The lack of precipitation (especially in summer)

may lead to a unimodal distributed MMD and ventilation condition in Urumqi. Apart from its one-peak pattern, the transport wind speed in Urumqi is weaker than that in Harbin and Beijing, owing to the block of Qinghai–Tibet Plateau in the south. Chongqing, on the other hand, is an extreme example, where atmospheric ventilation is poor (lower than 10000 $m^2/s$) throughout the year. Along with this are low MMD (lower than 1500 m) and weak transport wind (weaker than 4 m/s). This natural situation may put large pressure on local air quality.

### 3.2 Precipitation climatology

Another component for new ASI is precipitation, which is retained from the original metric for ASI. Figure 5 shows that the rainfall amount and rainy days are observed to be more significant over the southeast of China than inland areas due to the transport of moisture from the ocean by summer monsoon. Rainfall amount and rainy days of four

representative stations (Harbin, Urumqi, Beijing and Chongqing) are shown in Fig. 6. It is seen that generally,





Chongqing experiences the largest annual rainfall amount and Urumqi the least (Fig. 6a). Seasonally, rainfall is abundant in the summer monsoon season and scarce in winter for Harbin, Beijing and Chongqing. But for Urumqi, it keeps a low value throughout the year, illustrating the arid climate in the northwest of China. As for the rainy days, Harbin and Beijing experience the most of them in summer monsoon season, sharing the similar unimodal

pattern with their monthly rainfall amount. Rainy days of Urumqi keep a low value all the year round, the same as the pattern of its rainfall amount. However, the variation of rainy days (Fig. 6b) of Chongqing shows two distinct peaks: one is seen in May and June (about 12 days) and another in October (about 11 days). It indicates that autumn in Chongqing is characterized by moderate rain with more frequency.

**3.3 CAPE and CIN climatology**

Apart from the actual rainfall to washout air pollutants, CAPE values are also important since it relates to convection initiations and thermodynamic speed limit (Markowski and Richardson, 2011). Larger values of CAPE with CIN deducted (referred to as CAPE_CIN hereinafter) mean a higher probability in the destabilization of the atmosphere and genesis of stronger wet convection.

Annual mean values of CAPE (Fig. 7a) generally increase from south to north, following the spatial distribution of the near-surface specific humidity and temperature (Riemann-Campe et al., 2009 and their Fig. 2). The maxima occur in the southern coastal areas (larger than 300 J/kg), where the surface temperature is high and moisture is abundant. There is also large CAPE stored in the atmosphere over the southern region of Qinghai–Tibet Plateau.

As the summer monsoon of south Asia is established, warm moist air originating from the Arabian Sea and the Bay of Bengal is blown to Tibet plateau (Romatschke et at., 2010), and hence large value of CAPE exists. The minimal CAPE values are observed in the north especially the northwest i.e., the arid and relatively cold regions, where the value of CAPE is less than 30 J/kg. In contrast to relatively large values of CAPE, values of CIN over China range from 4 to 70 J/kg (Fig. 7b). Smaller values of CIN occur in the same region where smaller CAPE values are

observed. Large CIN values are observed in Sichuan basin, indicating the need of stronger forcing to overcome stable atmospheric layer for the occurrence of a wet convection. As a variable indicating real latent instability, annual mean CAPE_CIN (Fig. 7c) is spatially distributed similarly to CAPE. In general, smaller values are observed in the northwest of China (less than 10 J/kg) and larger values occur in the southmost part of the country and south of Qinghai–Tibet Plateau (larger than 300 J/kg).

The annual cycles of CAPE and CIN of four representative stations (Harbin, Urumqi, Beijing and Chongqing) are investigated (Fig. 8). CAPE values present a unimodal distribution. They generally increase in spring, achieve maxima in summer months, then begin to decrease during autumn season and reach minima in winter. The annual cycle of CIN of Harbin, Beijing and Chongqing displays a similar distribution as that of CAPE but not as

pronounced. As a result, CAPE of these three stations are much larger than the corresponding CIN during spring, summer and autumn. Chongqing presents the largest value of CAPE and CAPE_CIN, indicating the largest probability to form cumulus convection. In contrast, CIN value of Urumqi is comparable to its CAPE. Both of them are very small, ranging from 0 to 50 J/kg. In addition, the value of CIN is less than or almost equal to CAPE in autumn and winter months. Under such circumstances, destabilization will not occur, even if the layer has been

lifted sufficiently.

## 4. Results of new ASI

The national distribution of annual mean new ASI during 1985–2014 is displayed in Fig. 9. Results show that the spatial distributions of new air stagnation days and cases are basically consistent with the old ones displayed in

Huang et al. (2017). The new stagnation days and cases are still prevalent over Xinjiang and Sichuan basins (more than 180 days and 14 cases per year) and barely occur over Qinghai–Tibet and Yunnan plateaus (less than 40 days and 2 cases per year). However, the distribution pattern of the duration of one stagnation case is changed under new criteria. Air stagnation cases persist longer in the north part of China (more than 8 days) and then the central part (about 7 days). The longest duration is observed in the northwest region of China (more than 10 days), instead

of the south of this country in the earlier study (Huang et al., 2017). The shortest duration occurs over Qinghai–Tibet and Yunnan plateaus (less than 5 days).

Much improved are the seasonal cycles of air stagnation days and cases. Results of monthly variations of four representative stations (Harbin, Urumqi, Beijing and Chongqing), as well as the national averages, are displayed in

Fig. 10. In general, large ASI are observed in winter. In contrast to the original peak of ASI, air stagnation with the new standards is a minimum in summer, due to the relatively good ventilation condition, abundant rainfall, and higher probability to form cumulus convection in this season.



## 5. Correlations between the new ASI and actual air pollution

API as an indicator of air quality is used to testify the reasonability of the new ASI. Figure 11 shows that, monthly variations of the new ASI can basically represent the annual cycles of API. For all the four representative stations, the annual cycle of new air stagnation days reach maxima in winter and minima in summer, in agreement with the

annual cycle of API. However, there are two failures for this new ASI to predict the API. One is the secondary peak of API in spring for both Beijing and Harbin, the new ASI cannot represent it. This is attributed to the influence of sand storms in spring in northern China. Sand storms usually result in high level of particulate air pollution (high API), and are accompanied by strong winds and good ventilation conditions (low ASI). Another discrepancy is the high ASI in October and November in Urumqi, corresponding to relatively lower API values. We cannot explain

this discrepancy till now. The original ASI is also shown in Fig. 11. Obviously, the new ASI performs much better in indicating actual air pollution levels.

The correlations between monthly mean API and the total air stagnation days in this month are also investigated (Fig. 12). In order to exclude the influences of emissions as much as possible, the investigation only covers data of

winter half-year (i.e., October–March) when domestic heating requires more energy consumption. Figure 12 shows that with the old criteria, monthly mean API has no evident correlation with monthly stagnation days, or even decrease with them (Chongqing), which is clearly irrational. Instead, with the new criteria, monthly mean API basically increase with the number of air stagnation days as we expected.

In addition, we have also explored the ability of the new ASI to capture the air pollution days (Table 1). Every day during 2001–2012 is classified according to its API value. Then we calculated how many days in one category are considered as air stagnation day, under the original and the newly defined criteria. Results show that for moderate air quality with API ranging from 50 to 100, the number of stagnation days that identified by the new ASI is almost the same as or even less than that by the original one. For unhealthy air quality with API ranging from 101 to 500,

the percentage of identified air stagnation days with the new ASI is basically more than that with the original one, the former may even double or triple the latter in some situation. It suggests that the newly defined ASI has reduced the probability of misjudging an air pollution day and increased the chances to capture the true one.

Apart from the statistical analysis of correlations between API and ASI, we also conducted a case study about $PM_{2.5}$

concentrations and ASI (Fig. 13). It is found that the newly defined ASI is able to track the day-by-day variation of air pollution episode in January 2013 in Beijing, which was characterized by continuous high $PM_{2.5}$ concentrations

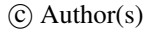


(Ji et al., 2014; Sun et al., 2014). The mean daily $PM_{2.5}$ concentration during that month ranged from 15.8 (January

1) to 568.6 (January 12) $\mu g / m^3$. There were only three days when the $PM_{2.5}$ concentration reached the standard

of less than 25 $\mu g / m^3$ for 24 h average suggested by the Air Quality Guidelines (AQG) of the World Health

Organization (WHO; WHO 2005), and only seven days met the China National Ambient Air Quality Standards

(CNAAQS; 75 $\mu g / m^3$ for 24 h average). In particular, the record-breaking hourly concentration of $PM_{2.5}$

reached as high as 886 $\mu g / m^3$ at 1900 local time on January 12, exceeding more than 11 times the CNAAQS

and 35 times the WHO AQG.

Ordinarily, emissions from one particular region remain constant over a short period. Therefore, the day-by-day

variation of local air quality mainly depends on its weather conditions. Hourly concentration of $PM_{2.5}$ of Beijing

and air stagnation days during January 2013 are presented in Fig. 13, as well as the components of both old and

new ASI. According to weather records, no precipitation is reported in this month except on January 20 and 31. In

addition, the values of CAPE and CIN are nearly zero in winter season, as mentioned above. Therefore, we only

focus on the ventilation condition of the new ASI, and 500 hPa and 10 m wind speeds of the old ASI. It is seen that

there are two long-duration haze episodes, i.e. January 10–14 and 25–30, and some minor $PM_{2.5}$ concentration

peaks in January 6–7, 16, 18–19, and 21–23 during this month. The old air stagnation metric only successfully

captures one heavy pollution in January 29 and one moderate pollution in January 21. But it fails to capture other

heavily polluted days, and misjudges a clean day (January 3) as a pollutant accumulation day. The reason is that

although 10 m wind speed falls below the 3.2 m/s standard since January 3, 500 hPa wind speed—the dominant

component in the original ASI (Huang et al., 2017)—is generally above the 13 m/s standard except for January 3,

21 and 29. It verifies again that 500 hPa wind speed is not appropriate to be an ASI element. By using our redefined

the ASI, the air stagnation pattern matches the day-by-day variation of $PM_{2.5}$ concentration pretty well. The two

major and four minor haze episodes are all successfully identified as stagnation day except January 11. It should

be reasonable since even though the daily mean concentration of $PM_{2.5}$ on January 11 was relatively high, its hourly

25   concentration actually decreased throughout the day. Correspondingly, this day presents a good ventilation

condition (9328 $m^2/s$; Fig. 13b) and is identified as a non-stagnation day.

**6. Conclusion**

Meteorological background is critical to the occurrence of air pollution events, since it determines the accumulation

or dilution of air pollutants. The United States adopts ASI to measure the meteorological state, taking 500 hPa and 10 m wind speeds, and precipitation into consideration. However, this metric is found not well adequate for China. Taking Beijing as an example, strong upper-level wind occurs in 75% of most polluted days (API ranks in the 90[th] percentile or higher). It means that strong upper-air winds does not necessarily drive the near surface dispersion of

air pollutants. In the present study, we proposed a new metric for ASI, based on (i) daily maximal ventilation, indicating horizontal dilution and vertical mixing extent in the boundary layer, (ii) real latent instability, implying the convection between the layer of cloud base and cloud top, (iii) precipitation, representing the washout effects of air pollutants.

Based on sounding data and hourly near-surface temperature observations, we firstly calculated daily maximum mixing layer depth during 1985–2014, and then estimated daily maximal ventilation in the boundary layer. Combined with daily precipitation and CAPE and CIN values, we derived results of the newly defined ASI of 66 stations across the country, and investigated the climatological features of them. It shows that the newly defined ASI keeps a similar spatial pattern in comparison with the original one (Huang et al., 2017). Annual mean

stagnation days under new metric are most prevalent (more than 180 days) over the northwest and southwest basins, i.e., Xinjiang and Sichuan basins, and least (less than 40 days) over Qinghai–Tibet and Yunnan plateaus. However, the annual cycle of the newly derived stagnation days is different, with the frequency of stagnation occurrences reaching a maximum in winter and minimum in summer, which is contrary to the original ASI result.

The annual cycle of new stagnation days is in consistent with the monthly variation of API during 2000–2012. Moreover, the new ASI shows a positive correlation with API during boreal winter half-year, and is found to be more efficient to identify unhealthy air quality with API ranging from 101 to 500. In addition, the new ASI also well track the day-by-day variation of $PM_{2.5}$ concentration during the record-breaking haze episodes in January 2013 in Beijing. Therefore, the new ASI may act as a better indicator of atmospheric dispersion and dilution in air

pollution research.

**Acknowledgements**

This work is partially supported by National Natural Science Foundation of China (41575007, 91544216) and Clean Air Research Project in China (201509001). We greatly acknowledge the work of University of Wyoming,

U.S.NCDC and CMA for providing long-term sounding data and surface observations, and also acknowledge the





Ministry of Environment Protection of China and the US Embassy for providing API and $PM_{2.5}$ concentration data, respectively.

**Competing interests**

5    The authors declare that they have no conflict of interest.

**Data availability**

Radiosonde data during 1985–2014 were obtained from the University of Wyoming (http://weather.uwyo.edu/upperair/sounding.html). Hourly observations of surface air temperature during 30 years

10    were extracted from ISD, provided by NCDC (ftp://ftp.ncdc.noaa.gov/pub/data/noaa). Dataset of daily precipitation during these 30 years were obtained from CMA website (http://data.cma.cn/data/detail/dataCode/SURF_CLI_CHN_MUL_DAY_CES_V3.0.html). API data during 2000–2012 were collected from Ministry of Environment Protection of the People's Republic of China (http://datacenter.mep.gov.cn). Hourly concentration data of $PM_{2.5}$ of Beijing in January of 2013 were collected

15    from the US Embassy (http://www.stateair.net/web/historical/1/1.html).





**Appendix A**

Table A1. The information of 66 stations used in this study. Date ranges are the periods when radiosonde data, daily precipitation and daily maximum temperature are all available at the same time. The periods less than 30 years are highlighted in bold. The percentages of valid data are presented.

| ID[*] | Name | Latitude (° N) | Longitude (° E) | Elevation (m) | Date Range | Valid Data |
|---|---|---|---|---|---|---|
| 50527 | Hailar | 49.21 | 119.75 | 611 | 198501-201412 | 97.79% |
| 50557 | Nenjiang | 49.16 | 125.23 | 243 | 198501-201412 | 98.88% |
| 50953 | Harbin | 45.75 | 126.76 | 143 | 198501-201412 | 99.34% |
| 51076 | Altay | 47.72 | 88.08 | 737 | 198501-201412 | 99.18% |
| 51431 | Yining | 43.95 | 81.33 | 664 | 198501-201412 | 99.26% |
| 51463 | Urumqi | 43.77 | 87.62 | 919 | 198501-201412 | 99.15% |
| 51644 | Kuqa | 41.71 | 82.95 | 1100 | 198501-201412 | 99.32% |
| 51709 | Kashi | 39.46 | 75.98 | 1291 | 198501-201412 | 99.17% |
| 51777 | Ruoqiang | 39.02 | 88.16 | 889 | 198501-201412 | 99.09% |
| 51828 | Hotan | 37.13 | 79.93 | 1375 | 198501-201412 | 99.34% |
| 52203 | Hami | 42.81 | 93.51 | 739 | 198501-201412 | 99.27% |
| 52418 | Dunhuang | 40.15 | 94.68 | 1140 | 198501-201412 | 99.05% |
| 52533 | Jiuquan | 39.75 | 98.48 | 1478 | 198501-201412 | 98.87% |
| 52681 | Minqin | 38.63 | 103.08 | 1367 | 198501-201412 | 98.46% |
| 52818 | Golmud | 36.4 | 94.9 | 2809 | 198501-201412 | 99.17% |
| 52836 | Dulan | 36.29 | 98.09 | 3192 | 198501-201412 | 96.88% |
| 52866 | Xining | 36.71 | 101.75 | 2296 | 198501-201412 | 99.04% |
| 53068 | Erenhot | 43.65 | 112 | 966 | 198501-201412 | 99.02% |
| 53463 | Hohhot | 40.81 | 111.68 | 1065 | 198501-201412 | 99.25% |
| 53614 | Yinchuan | 38.47 | 106.2 | 1112 | 198501-201412 | 98.17% |
| 53772 | Taiyuan | 37.77 | 112.55 | 779 | 198501-201412 | 98.96% |
| 53845 | YanAn | 36.59 | 109.5 | 959 | 198501-201412 | 99.32% |
| 53915 | Pingliang | 35.54 | 106.66 | 1348 | 198501-201412 | 98.88% |
| 54102 | XilinHot | 43.95 | 116.05 | 991 | 198501-201412 | 99.20% |
| 54135 | Tongliao | 43.6 | 122.26 | 180 | 198501-201412 | 99.25% |
| 54161 | Changchun | 43.9 | 125.21 | 238 | 198501-201412 | 98.98% |
| 54218 | Chifeng | 42.25 | 118.95 | 572 | 198501-201412 | 99.26% |
| 54292 | Yanji | 42.88 | 129.46 | 178 | 198501-201412 | 99.32% |
| 54342 | Shenyang | 41.75 | 123.43 | 43 | 198501-201412 | 99.16% |
| 54374 | Linjiang | 41.71 | 126.91 | 333 | 198501-201412 | 98.84% |
| 54511 | Beijing | 39.93 | 116.28 | 55 | 198501-201412 | 99.25% |





| ID* | Name | Latitude (° N) | Longitude (° E) | Elevation (m) | Date Range | Valid Data |
|---|---|---|---|---|---|---|
| 54662 | Dalian | 38.9 | 121.62 | 97 | 198501-201412 | 99.16% |
| 55591 | Lhasa | 29.65 | 91.12 | 3650 | 198501-201412 | 97.72% |
| 56029 | Yushu | 33 | 97.01 | 3682 | 198501-201412 | 97.11% |
| 56080 | Hezuo | 35 | 102.9 | 2910 | 198501-201412 | 98.39% |
| 56146 | Garze | 31.61 | 100 | 522 | 198501-201412 | 98.64% |
| 56187 | Wenjiang | 30.7 | 103.83 | 541 | **200501-201412** | 98.03% |
| 56571 | Xichang | 27.9 | 102.26 | 1599 | 198501-201412 | 98.65% |
| 56739 | Tengchong | 25.11 | 98.48 | 1649 | 198501-201412 | 99.02% |
| 56778 | Kunming | 25.01 | 102.68 | 1892 | 198501-201412 | 98.99% |
| 56964 | Simao | 22.76 | 100.98 | 1303 | 198501-201412 | 99.24% |
| 56985 | Mengzi | 23.38 | 103.38 | 1302 | 198501-201412 | 99.22% |
| 57083 | Zhengzhou | 34.7 | 113.65 | 111 | 198501-201412 | 99.31% |
| 57127 | Hanzhong | 33.06 | 107.02 | 509 | 198501-201412 | 99.34% |
| 57131 | Jinghe | 34.26 | 108.58 | 411 | **200710-201412** | 99.92% |
| 57447 | Enshi | 30.28 | 109.46 | 458 | 198501-201412 | 99.33% |
| 57461 | Yichang | 30.7 | 111.3 | 134 | 198501-201412 | 99.31% |
| 57494 | Wuhan | 30.61 | 114.12 | 23 | 198501-201412 | 99.32% |
| 57516 | Chongqing | 29.51 | 106.48 | 260 | **198708-201412** | 98.56% |
| 57816 | Guiyang | 26.47 | 106.65 | 1222 | 198501-201412 | 98.57% |
| 57957 | Guilin | 25.33 | 110.3 | 166 | 198501-201412 | 99.33% |
| 57993 | Ganzhou | 25.85 | 114.94 | 125 | 198501-201412 | 99.27% |
| 58027 | Xuzhou | 34.27 | 117.15 | 42 | 198501-201412 | 99.33% |
| 58238 | Nanjing | 32 | 118.8 | 7 | 198501-201412 | 99.34% |
| 58362 | Shanghai | 31.4 | 121.46 | 4 | **199106-201412** | 98.88% |
| 58424 | Anqing | 30.53 | 117.05 | 20 | 198501-201412 | 99.33% |
| 58457 | Hangzhou | 30.22 | 120.16 | 43 | 198501-201412 | 99.05% |
| 58606 | Nanchang | 28.6 | 115.91 | 50 | 198501-201412 | 99.31% |
| 58633 | QuXian | 28.95 | 118.86 | 71 | 198501-201412 | 99.03% |
| 58847 | Fuzhou | 26.07 | 119.27 | 85 | 198501-201412 | 99.24% |
| 59134 | Xiamen | 24.47 | 118.08 | 139 | 198501-201412 | 99.29% |
| 59211 | Baise | 23.9 | 106.6 | 175 | 198501-201412 | 99.15% |
| 59265 | Wuzhou | 23.48 | 111.3 | 120 | 198501-201412 | 98.29% |
| 59316 | Shantou | 23.35 | 116.66 | 3 | 198501-201412 | 99.23% |
| 59431 | Nanning | 22.62 | 108.2 | 126 | 198501-201412 | 99.28% |
| 59758 | Haikou | 20.03 | 110.34 | 24 | 198501-201412 | 99.13% |

* World Meteorological Organization Identification Number.



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





**Figure Captions**

Figure 1. The relationship between air pollution index (API) and 500 hPa wind speeds in most polluted days (i.e., days when API ranks in the 90$^{th}$ percentile or higher) in boreal winter half-year (October–March) during 2000–2012 of Beijing station. Air pollutions are classified into six levels (I–VI) which means excellent, good, lightly

polluted, moderately polluted, heavily polluted, and severely polluted conditions, respectively.

Figure 2. Topography of China and locations of 66 stations analyzed in this study. Cross marks indicate the four representative stations: Harbin, Urumqi, Beijing and Chongqing.

Figure 3. The spatial distribution of daily maximal ventilation (a), MMD (b) and transport wind speed (c) averaged over 30 years (1985–2014).

Figure 4. Annual cycle of ventilation, MMD and transport wind speed of four representative stations (Harbin, Urumqi, Beijing and Chongqing). Monthly mean values are given as horizontal bars in the middle, 25% and 75%

percentiles are shown as boxes' lower and upper boundaries, and 10% and 90% percentiles as lower and upper whiskers.

Figure 5. The spatial distribution of annual precipitation (a) and precipitation days (b) during 1985–2014.

Figure 6. Annual cycle of precipitation (a) and precipitation days (b) of four representative stations (Harbin, Urumqi, Beijing and Chongqing).

Figure 7. The spatial distribution of annual mean CAPE (a), CIN (b) and real latent instability (c) during 1985–2014.

Figure 8. Annual cycle of CAPE (black line) and CIN (green line) values of four representative stations (Harbin, Urumqi, Beijing and Chongqing). The orange filled area indicates values of CAPE with CIN deducted, i.e., real latent instability.

Figure 9. The spatial distribution of annual mean stagnation days (a), stagnation cases (b) and duration of stagnation cases (c) under new metrics.





Figure 10. Annual cycles of air stagnation days and cases of four representative stations (Harbin, Urumqi, Beijing and Chongqing), as well as the national average values. Black and grey lines indicate stagnation conditions under new and original metrics, respectively. Solid and dash lines represents stagnation days and cases, respectively.

Figure 11. Monthly variations of API averaged over 13 years (2000–2012), as well as the corresponding air stagnation days under the original and new criteria. Blue line: the original ASI; red line: the new ASI.

Figure 12. Correlations between monthly mean API and the total air stagnation days in the corresponding month in China winter half-year (October–March) during 2000–2012. Blue circles: original ASI; red dots: new ASI.

Figure 13. Time series of ASI and hourly concentration of PM$_{2.5}$ during January 2013 of Beijing (a), and the corresponding variation of main components of new and old ASI, i.e., daily ventilation and 500 hPa and 10 m wind speeds (b).



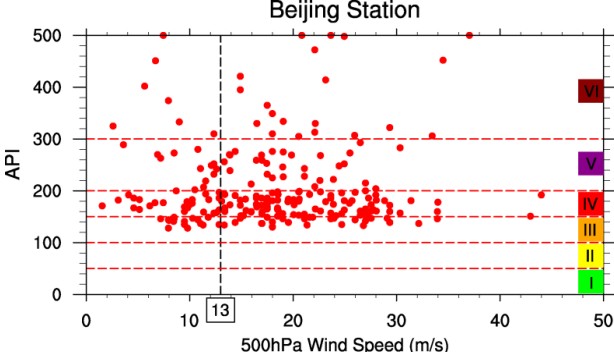

**Figure 1. The relationship between air pollution index (API) and 500 hPa wind speeds in most polluted days (i.e., days when API ranks in the 90th percentile or higher) in boreal winter half-year (October–March) during 2000–2012 of Beijing station. Air pollutions are classified into six levels (I–VI) which means excellent, good, lightly polluted, moderately polluted, heavily polluted, and severely polluted conditions, respectively.**




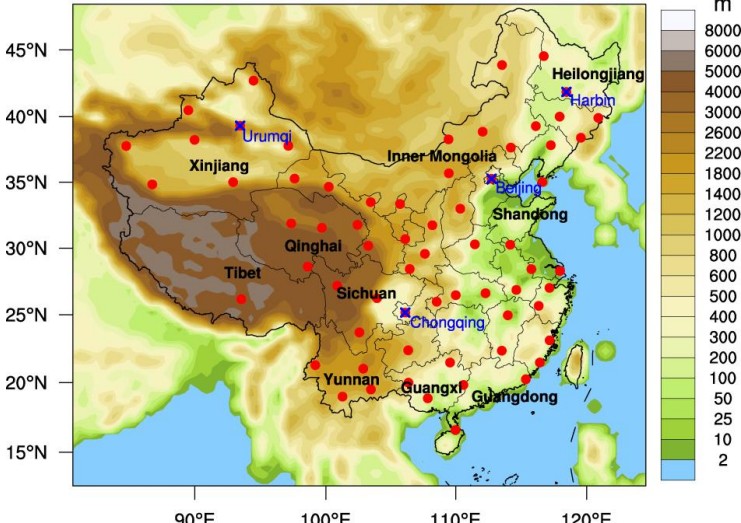

**Figure 2. Topography of China and locations of 66 stations analyzed in this study. Cross marks indicate the four representative stations: Harbin, Urumqi, Beijing and Chongqing.**



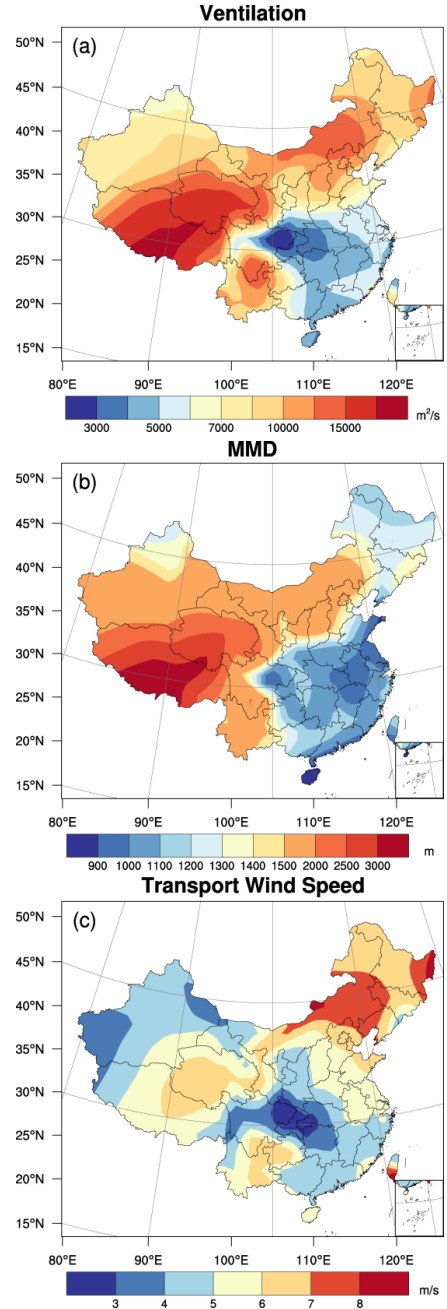

**Figure 3. The spatial distribution of daily maximal ventilation (a), MMD (b) and transport wind speed (c) averaged over 30 years (1985–2014).**

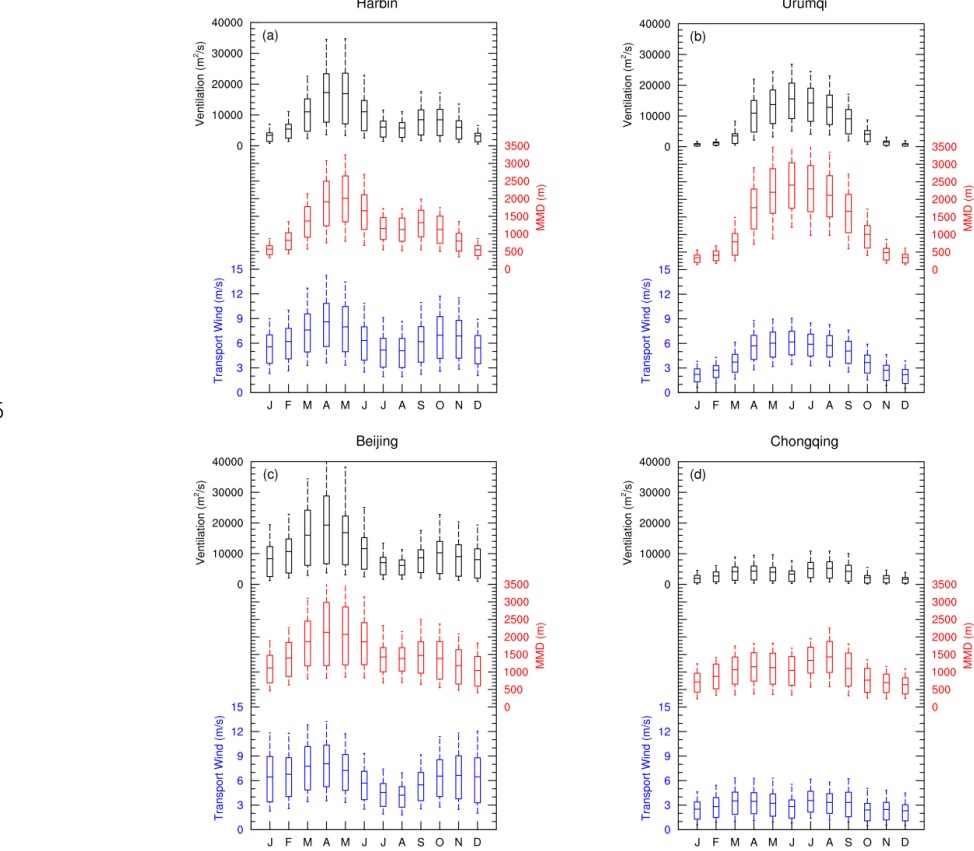

**Figure 4. Annual cycle of ventilation, MMD and transport wind speed of four representative stations (Harbin, Urumqi, Beijing and Chongqing). Monthly mean values are given as horizontal bars in the middle, 25% and 75% percentiles are shown as boxes' lower and upper boundaries, and 10% and 90% percentiles as lower and upper whiskers.**




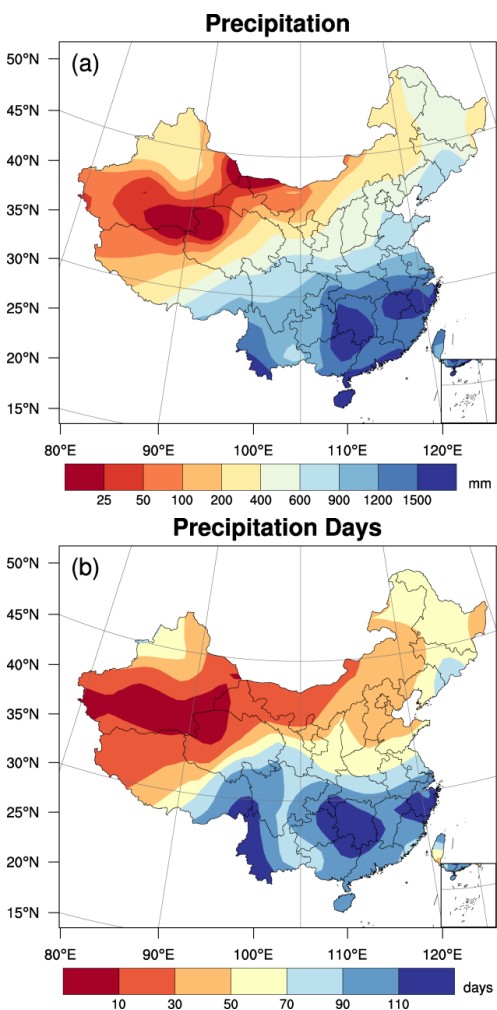

**Figure 5. The spatial distribution of annual precipitation (a) and precipitation days (b) during 1985–2014.**



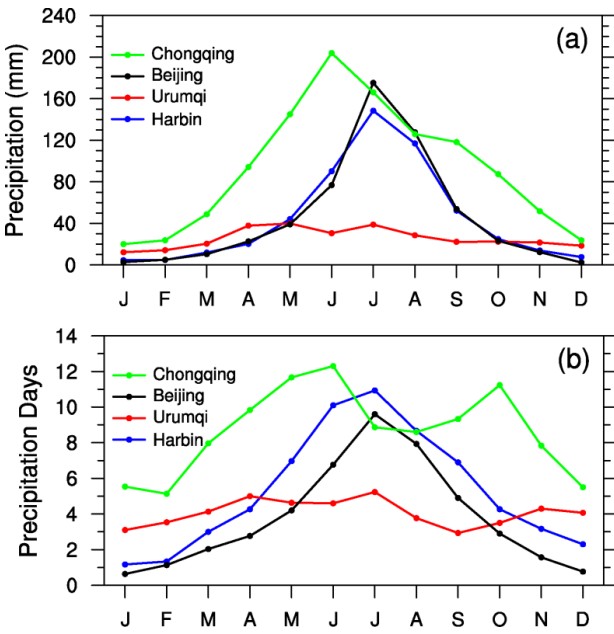

**Figure 6. Annual cycle of precipitation (a) and precipitation days (b) of four representative stations (Harbin,**

10    **Urumqi, Beijing and Chongqing).**





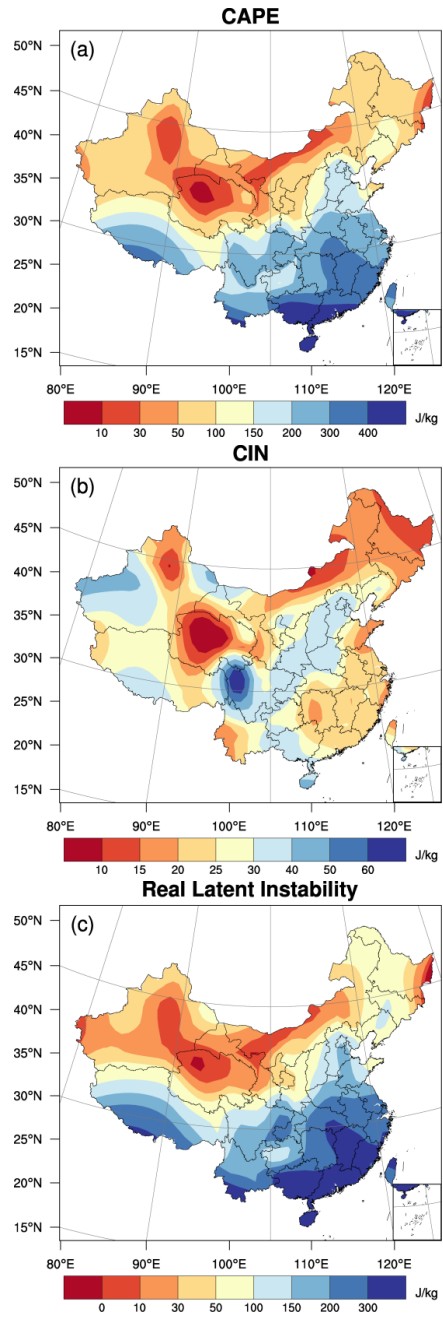

**Figure 7. The spatial distribution of annual mean CAPE (a), CIN (b) and real latent instability (c) during 1985–2014.**



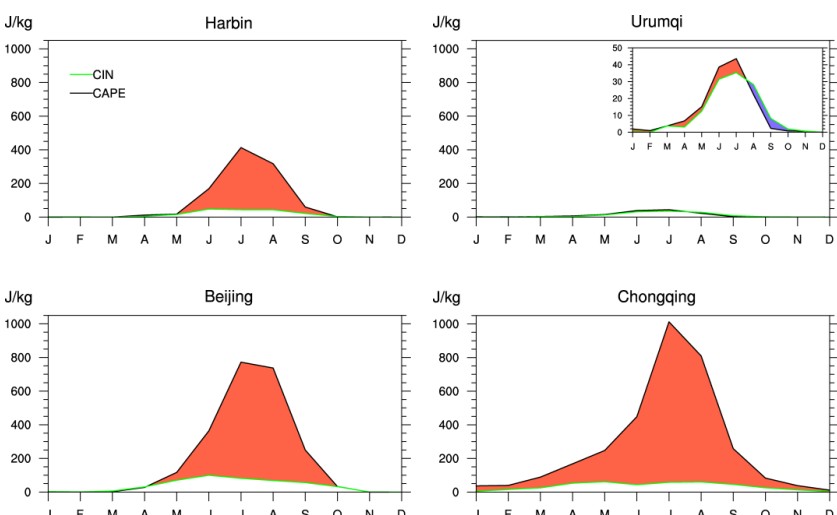

**Figure 8. Annual cycle of CAPE (black line) and CIN (green line) values of four representative stations**

10 **(Harbin, Urumqi, Beijing and Chongqing). The orange filled area indicates values of CAPE with CIN**

**deducted, i.e., real latent instability.**





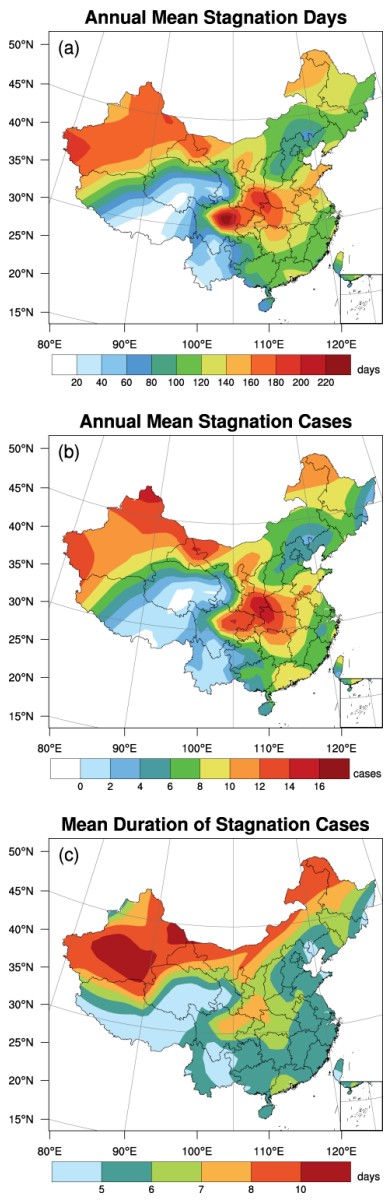

**Figure 9. The spatial distribution of annual mean stagnation days (a), stagnation cases (b) and duration of stagnation cases (c) under new metrics.**





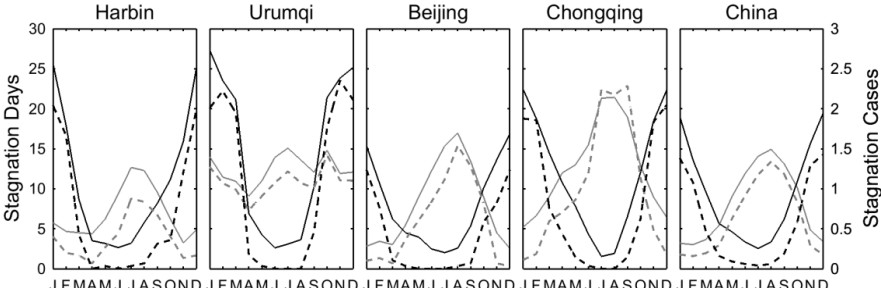

**Figure 10. Annual cycles of air stagnation days and cases of four representative stations (Harbin, Urumqi,**
**Beijing and Chongqing), as well as the national average values. Black and grey lines indicate stagnation**
**conditions under new and original metrics, respectively. Solid and dash lines represents stagnation days and**
**cases, respectively.**



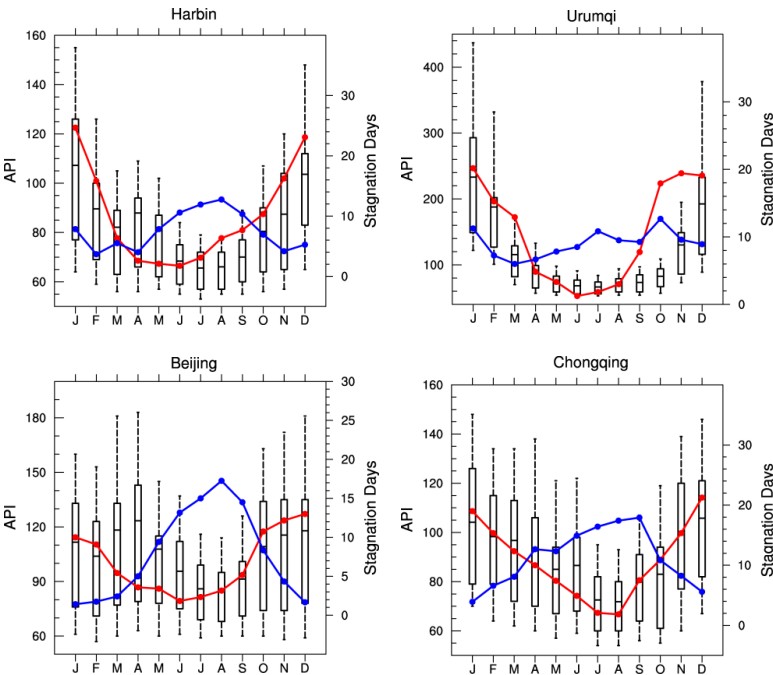

**Figure 11. Monthly variations of API averaged over 13 years (2000–2012), as well as the corresponding air**

10  **stagnation days under the original and new criteria. Blue line: the original ASI; red line: the new ASI.**



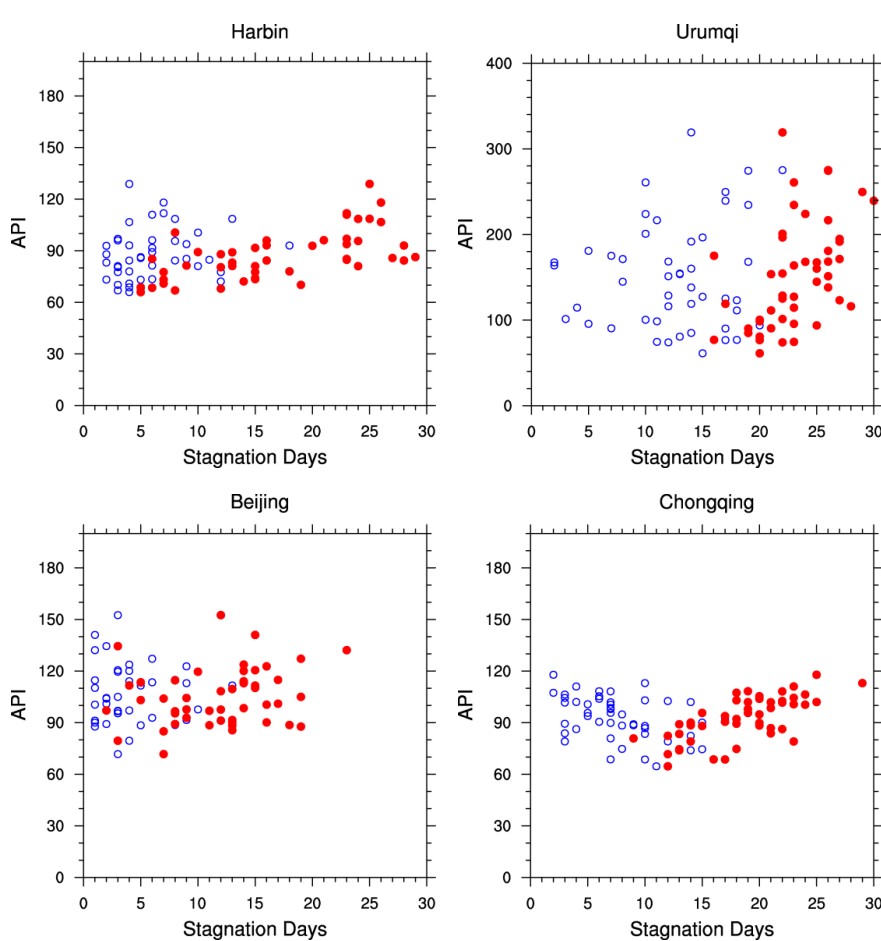

**Figure 12. Correlations between monthly mean API and the total air stagnation days in the corresponding month in China winter half-year (October–March) during 2000–2012. Blue circles: original ASI; red dots: new ASI.**





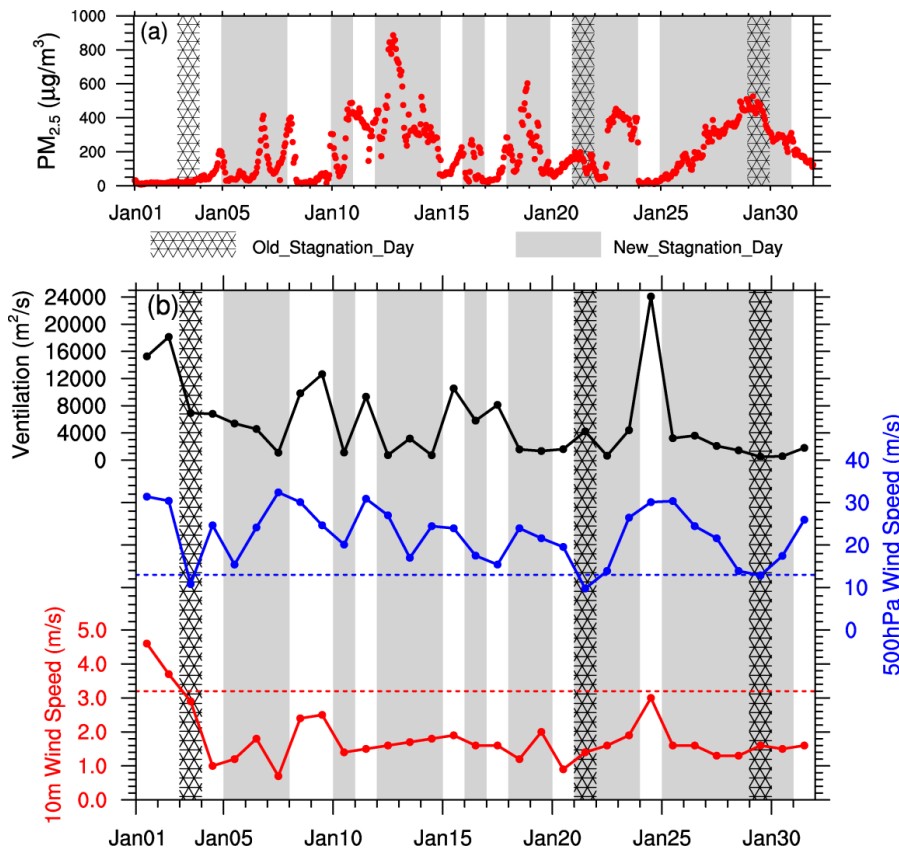

**Figure 13. Time series of ASI and hourly concentration of PM$_{2.5}$ during January 2013 of Beijing (a), and the corresponding variation of main components of new and old ASI, i.e., daily ventilation and 500 hPa and 10 m wind speeds (b).**





**Table**

**Table 1. How many days are identified as air stagnation days in each API category?**

| API Classification | Harbin | | Urumqi | | Beijing | | Chongqing | |
|---|---|---|---|---|---|---|---|---|
| | New ASI | Old ASI | New ASI | Old ASI | New ASI | Old ASI | New ASI | Old ASI |
| 0-50 | ---[a] | --- | --- | --- | --- | --- | --- | --- |
| 51-100 | 30.5% | 28.1% | 31.1% | 39.7% | 20.1% | 31.6% | 37.1% | 45.7% |
| 101-150 | 64.4% | 24.6% | 79.4% | 35.7% | 35.2% | 34.3% | 59.7% | 41.9% |
| 151-200 | 87.2% | 23.4% | 87.9% | 42.8% | 45.4% | 27.5% | 64.9% | 52.7% |
| 201-300 | 70.0% | 30.0% | 91.8% | 47.5% | 49.3% | 16.4% | 50.0% | 25.0% |
| 301-500 | 28.6% | 28.6% | 89.9% | 56.9% | 41.4% | 13.8% | --- | --- |

[a] The result is ignored because there are less than 5 days falling into this category.