# Peer review of "Climatological study of boundary-layer air stagnation index for China and its relationship with air pollution"

_Atmospheric Chemistry and Physics, 2017_

## Referee Comment (RC1) · Anonymous Referee #1 · 23 Jan 2018

Review of "Climatological study of a new air stagnation index (ASI) for China and its relationship with air pollution" by Huang et al. (MS ID: #ACP-2017-1145)

Summary: This study has improved the ability of an air stagnation index to measure the atmospheric conditions of the air pollutions over China. The result is valuable and interesting and the paper is well written. I think this manuscript can meet the scope of ACP. I recommended it to be published in ACP after the following issues addressed clearly.

Specific Comments: 1. As indicated in the stage of quick comment, I have pointed that this study was quite similar with a newly published paper in the journal of "Bulletin

of the American Meteorological Society" (Wang et al. [kcwang@bnu.edu.cn], PM2.5 pollution in China and how it has been exacerbated by terrain and meteorological conditions, BAMS), at least, the definition of the air stagnation index. However, I have not got the available information related to the difference between them, though authors have cited this newly study. So, I still suggest the authors clarify it in the introduction. 2. It has been also pointed out in the quick comment but no reflection has been reached. The hourly PM2.5 data of Beijing in January 2013 from the US Embassy is used here. However, in general, this sort of data cannot be used in the open published paper because this monitoring is not a regularly site-observation. 3. Some newly works in this aspect should be reviewed. For example: (1) Cai WJ et al., 2017: Weather conditions conducive to Beijing severe haze more frequent under climate change. Nature Climate Change, doi:10.1038/NCLIMATE3249. (2) Yin ZC et al., 2017: Understanding severe winter haze events in the North China Plain in 2014: roles of climate anomalies. Atmos. Chem. Phys., 17, 1641-1651. (3) Han ZY et al., 2017: Projected changes in haze pollution potential in China: an ensemble of regional climate model simulations. Atmos. Chem. Phys., 17, 10109-10123. (4) Wang HJ et al., 2016: Understanding the recent trend of haze pollution in eastern China: roles of climate change. Atmos. Chem. Phys., 16, 4205-4211. 4. Some methodology explanations should be added. For example, "temperature profiles from radiosonde are linearly interpolated to 1-m vertical intervals", "Wind profile from 1200 UTC (i.e., 2000 BJT) sounding data is also interpolated to 1-m vertical grids,"...... how to complete it? Different results may be obtained when different methods used. 5. In this study, just 66 stations are used across China which is resampled into 2*2 grids. Obviously, there are many grids that even have no stations, resulting in misleading to readers for the information over these grids. Of course, some descriptions in this MS are not precise. For example, there is just one station in Tibet Plateau and the information of the spatial patterns for the ventilation, CAPE etc. are just the interpolation results, cannot represent the actual distribution. So, the authors say that the ventilation (CAPE etc.) is largest over Tibet Plateau may be not correct. 6. "Another discrepancy is the high ASI in October and November in

Urumqi, corresponding to relatively lower API values." I suggest the authors to check the variation of each component of ASI. May be it resulted by one of the components. 7. "In order to exclude the influences of emissions as much as possible, the investigation only covers data of winter half-year (i.e., October–March) when domestic heating requires more energy consumption." This sentence confused me. 8. In this study, the newly developed ASI is compared with the original one and the results indicating a better performance for newly index to capture the air stagnation days. The correlation coefficients should be shown in the text that can increase the readability. 9. From the comparison between the newly and original ASI, we can find that there are generally peak stagnation days in summer from original ASI but winter from newly one. Why? 10. From Figure 12, we can see that the numbers of stagnation days are generally much larger from the newly ASI than the original one. Why? 11. Some figure captions are not clear. Please check it. For example, what's the mean of the whisker in Figure 11.

---

## Referee Comment (RC2) · Anonymous Referee #2 · 4 Feb 2018

Synopsis: The authors design an atmospheric stagnation index for use in China, contrast it with the U.S. NCDC ASI, and suggest that theirs better captures the meteorological conditions conducive to poor AQ.

Recommendation: This is an interesting and needed study. The disconnect between AQ observations and the NCDC ASI over China has been shown in many previous publication. This, of course, has motivated many to design a better approach. In this attempt the authors propose meteorological factors that integrate scavenging, dispersal, and ventilation effects. I find the meteorological basis of the index design to be sound. However, I think its application and assessment require greater rigor, while the

manuscript itself could be better organized and written.

My editorial and scientific suggestions follow:

From a communication standpoint, I would advise the authors to clearly establish a new metric, i.e., do not call your index the "new ASI." The author's should look at this as an opportunity...an opportunity that will assist them in improving the clarity of the manuscript.

The creation of this index is ripe for multiple linear regression analysis. The authors suggest the ASI is overly reliant on 500mb winds. It seems likely that this new index may have similar issues. For example, if the CAPE/CIN is often zero, perhaps it provides little value to the calculation. With multiple linear regression analysis, one could quantitatively learn the value each new component adds to the overall result. I would advise this analysis for both monthly and daily data, where available.

From a fine temporal resolution perspective, I appreciate the focused analysis on Beijing, however I wonder why this portion of the analysis wasn't extended into other seasons and time periods. Does the "new ASI" perform as well during poor AQ events in the summer/spring/fall seasons? If daily Embassy data is sufficient to analyze in January of 2013, surely you could test your metric on other time periods.

Annual map plots, while instructive, are not particularly useful when discussing seasonally dependent meteorology. I would suggest presenting seasonal panels of your index components and discussing the seasonal variance in controlling drivers (multiple linear regression analysis would be helpful for this).

---

## Author Comment (AC1) · 16 May 2018

**Response to Referees**

**Manuscript: Climatological study of a new air stagnation index (ASI) for China and its relationship with air pollution (acp-2017-1145)**

We are very grateful to the referees for their careful and insightful comments. With their suggestions, this manuscript has been greatly improved. The referees' all comments are copied below in italics and followed by our responses. The corresponding modifications in manuscript are marked in blue color. Table 1 in the original manuscript has been renumbered as Table 2 in the revised version.

According to the suggestion of Referee #2, the newly defined stagnation index in our study is named BSI in the revision, short for Boundary-layer air Stagnation Index. Therefore, we use the name "BSI" instead of "the new index" in this response and the revision.

**Referee #1:**

*Summary: This study has improved the ability of an air stagnation index to measure the atmospheric conditions of the air pollutions over China. The result is valuable and interesting and the paper is well written. I think this manuscript can meet the scope of ACP. I recommended it to be published in ACP after the following issues addressed clearly.*

**Response:**

    We thank the referee for the positive evaluation to this article.

*Specific Comments:*

*1. As indicated in the stage of quick comment, I have pointed that this study was quite similar with a newly published paper in the journal of "Bulletin of the American Meteorological Society" (Wang et al. [kcwang@bnu.edu.cn], PM2.5 pollution in China and how it has been exacerbated by terrain and meteorological conditions, BAMS), at least, the definition of the air stagnation index. However, I have not got the available information related to the difference between them, though authors have cited this newly study. So, I still suggest the authors clarify it in the introduction.*

**Response:**

    Wang et al. (2017) also tried to redefine ASI to better describe the dispersion capacity of the atmosphere of China. In their definition, 10 m wind speed, boundary layer height and the occurrence of precipitation are taken into consideration. Their air stagnation threshold is determined by a fitting equation which relates to PM$_{2.5}$ concentration, wind speed and boundary layer height. This equation varies with locations and changes over time. In contrast, we avoid to relate the new index directly to air pollution monitoring data. Instead, we put the thresholds on meteorological basis

only. We hope that the proposed index is more universal and robust in principle. This goal is achieved, since the BSI shows its improved ability to indicate the annual cycles of air pollution levels (Fig. 11). Furthermore, we check the day-by-day variation of the BSI to PM$_{2.5}$ concentration through typical months in Beijing and it turns out that the BSI is able to track the daily variation of particulate matter (Figs. 13–15).

This response has been added in the Introduction.

References:

Wang, X., Dickinson, R. E., Su, L., Zhou, C., and Wang, K.: PM2.5 Pollution in China and How It Has Been Exacerbated by Terrain and Meteorological Conditions, Bulletin of the American Meteorological Society, doi:10.1175/bams-d-16-0301.1, 2017.

*2. It has been also pointed out in the quick comment but no reflection has been reached. The hourly PM2.5 data of Beijing in January 2013 from the US Embassy is used here. However, in general, this sort of data cannot be used in the open published paper because this monitoring is not a regularly site-observation.*

**Response:**

Hourly concentration of PM$_{2.5}$ data during January 2013 in Beijing are not yet accessible from Ministry of Environmental Protection (MEP) of China (http://datacenter.mep.gov.cn/). So we analyzed the data from US Embassy instead. According to Liang et al. (2016), PM$_{2.5}$ data from the US post and MEP stations in Beijing (i.e., Dongsi, Nongzhanguan, Dongsihuan, the nearest stations to the US Embassy station) were highly consistent. Moreover, data from US Embassy have already been used in many open published papers such as Cai et al. (2017) and Lv et al., (2017). Therefore, we considered PM$_{2.5}$ data from US Embassy be reliable. Besides, we have added more case studies of Beijing during 2015–2017 in this revision, using 24-h averaged PM$_{2.5}$ concentrations from MEP of China (Figs. 14 and 15).

**Response:**

Available observation data are indeed very limited over Qinghai–Tibet Plateau. Our results over this area are not very precise since the observation stations are sparse. Therefore, some of our descriptions need to be modified to be more modest. But in general, our results are reasonable, as explained below.

First, the interaction between the plateau and the atmosphere has long been recognized. High terrain elevation corresponds to thinner air and lower value of aerosol optical thickness, less precipitable water vapor amount, and stronger solar radiation (Chen et al., 2014; Zhu et al., 2010), which enhance thermal convections and lead to very high mixing layer over the plateau. The results agree with the recent study of Liu et al. (2015). On the other hand, the plateau locates right in the westerlies, and it can even break the jet stream into two branches surrounding the northern and southern edges of the plateau. Therefore, as Schiemann et al. (2009) revealed, the transport wind is strong (more than 6 m/s) over plateaus because of the combined influence of westerly jet and elevated topography. As a result, ventilation is largest over Tibet Plateau as a combined effect of the large mixing layer height and strong transport wind.

Second, there is also large CAPE in the atmosphere over the southern Qinghai–Tibet Plateau, when the summer monsoon of south Asia is established, with warm and moist air mass originating from the Bay of Bengal being blown to Tibet plateau (Romatschke et at., 2010).

We have rewritten related sentences in this revision.

**Response:**

According to the reviewer's suggestion, we check each component of the BSI.

As shown in Figs. 4, 6 and 8 in the manuscript, ventilation condition in Urumqi is good from April to September with average values larger than 10000 $m^2$/s, and becomes much worse from October to March in the next year with average values less than 5000 $m^2$/s. We counted days with poor ventilation conditions (i.e. daily maximum ventilation is less than 6000 $m^2$/s) in every month during 1985 to 2014 (Fig. R1), and found that poor ventilation conditions are rarely happen during April to September but occur frequently during October to March. The annual cycle of poor ventilation days matches well with that of BSI. The other two components CAPE_CIN and precipitation days keep a rather constant value yearly around (Figs. 6 and 8). So, we confirm that the poor ventilation is the dominant contributor for high BSI in October and November in Urumqi.

But we are still unable to explain the different trend between BSI and API during October to December in Urumqi: the ventilation condition is almost equally poor in these three months; whereas, the API grows significantly from October to December.

[Figure]

Figure R1. Annual cycle of poor ventilation days (i.e. days with ventilation less than 6000 m²/s).

*7. "In order to exclude the influences of emissions as much as possible, the investigation only covers data of winter half-year (i.e., October–March) when domestic heating requires more energy consumption." This sentence confused me.*
**Response:**

This sentence has been rewritten as follows:

"In order to exclude the impact of seasonal variation in source strengths, the investigation only covers data of winter half-year (i.e., October–March) when domestic heating in north China leads to more energy consumption and more serious air pollution pressure."

*8. In this study, the newly developed ASI is compared with the original one and the results indicating a better performance for newly index to capture the air stagnation days. The correlation coefficients should be shown in the text that can increase the readability.*
**Response:**

Accepted. The correlation coefficients have been added in Fig. 12.

*9. From the comparison between the newly and original ASI, we can find that there are generally peak stagnation days in summer from original ASI but winter from newly one. Why?*
**Response:**

In general, stagnation days under ASI metric is more in summer and less in winter. According to Huang et al. (2017), the behavior of upper-air wind speeds is the main driver of the ASI distribution. A weaker latitudinal pressure gradient at the upper layer of the atmosphere results in more air stagnation occurrences in summer. In contrast, the BSI is observed to be maximum in winter and minimum in summer. This behavior results from the cumulative responses of its components. We use standardized partial regression coefficients from multiple linear regression analysis to determine which variable is more important (Table R1, please see the answers to Questions 2 from Referee #2). It is shown that during the wintertime of Harbin, Urumqi and Beijing, ventilation is the main driver of the BSI; precipitation contributes less; and cumulus

convection barely plays a part. In summer, however, ventilation and cumulus convection both contribute a lot. For Harbin, they play an almost equal role, with regression coefficients of 0.57 and 0.48 respectively. For Beijing, the contribution of CAPE out weights that of ventilation more than twice. For Urumqi which is in the semi-arid climate region, the ventilation is still the main driver of the behavior of the BSI ($\beta$ =0.83), while precipitation and CAPE contribute equally ($\beta$ =0.36 and 0.4 respectively). The characteristics in Chongqing is different from the other three stations. With more convective energy stored in the atmosphere (Fig. 8), the seasonal variation of the BSI relies more on CAPE: it is the second contributor during winter and spring ($\beta$ =0.36 and 0.38 respectively) and becomes the main contributor in summer and autumn ($\beta$ =0.75 and 0.68 respectively). Noticeably in summer, the BSI is dominated by cumulus convection, while the ventilation barely plays a role.

The intention we add CAPE/CIN into the criteria is to make up for the deficiency of daily maximum mixing layer depth (MMD) derived from parcel method failing to describe the convection between the cloud base and cloud top. Therefore, it is expected that CAPE/CIN plays a more important role in summer.

This explanation has been added in Section 4.

**Response:**
Figure 12 only displayed stagnant conditions of winter half-year (i.e., October–March). The BSI is larger in winter and smaller in summer, while the ASI is larger in summer (Fig. 10). Therefore, when we compare the original and new stagnation index in the winter half-year, the numbers of stagnation days are generally much larger for the BSI. The explanation has been added in the manuscript.

*11. Some figure captions are not clear. Please check it. For example, what's the mean of the whisker in Figure 11.*
**Response:**
Monthly mean values of API are given as horizontal bars in the middle, 25% and 75% percentiles are shown as boxes' lower and upper boundaries, and 10% and 90% percentiles as lower and upper whiskers. This explanation has been added in the figure caption.

**Referee #2:**

*Synopsis: The authors design an atmospheric stagnation index for use in China, contrast it with the U.S. NCDC ASI, and suggest that theirs better captures the meteorological conditions conducive to poor AQ. Recommendation: This is an*

*interesting and needed study. The disconnect between AQ observations and the NCDC ASI over China has been shown in many previous publication. This, of course, has motivated many to design a better approach. In this attempt the authors propose meteorological factors that integrate scavenging, dispersal, and ventilation effects. I find the meteorological basis of the index design to be sound. However, I think its application and assessment require greater rigor, while the manuscript itself could be better organized and written.*

**Response:**

The referee's positive evaluation is very much appreciated.

*My editorial and scientific suggestions follow:*
*1. From a communication standpoint, I would advise the authors to clearly establish a new metric, i.e., do not call your index the "new ASI." The author's should look at this as an opportunity: : :an opportunity that will assist them in improving the clarity of the manuscript.*

**Response:**

We thank the referee for this advice. In the revised manuscript, the new ASI is called BSI, short for Boundary-layer air Stagnation Index.

*2. The creation of this index is ripe for multiple linear regression analysis. The authors suggest the ASI is overly reliant on 500mb winds. It seems likely that this new index may have similar issues. For example, if the CAPE/CIN is often zero, perhaps it provides little value to the calculation. With multiple linear regression analysis, one could quantitatively learn the value each new component adds to the overall result. I would advise this analysis for both monthly and daily data, where available.*

**Response:**

The intention we add CAPE/CIN into the criteria is to make up for the deficiency of daily maximum mixing layer depth (MMD) derived from parcel method failing to describe the convection between the cloud base and cloud top. Therefore, it is expected that CAPE/CIN plays a more important role in summer.

[revised manuscript text omitted]

*4. Annual map plots, while instructive, are not particularly useful when discussing seasonally dependent meteorology. I would suggest presenting seasonal panels of your*

*index components and discussing the seasonal variance in controlling drivers (multiple linear regression analysis would be helpful for this).*

**Response:**

Seasonal variation of the BSI and its components are added in the Supplement. The seasonally dependence of BSI on its components has been discussed using standardized partial regression coefficients from multiple linear regression analysis (Table R1). Please refer to the answers to Question 2. The analysis has been added in this revision.

---

## Author Comment (AC2) · 16 May 2018

**Response to Referees**

Manuscript: Climatological study of a new air stagnation index (ASI) for

China and its relationship with air pollution (acp-2017-1145)

We are very grateful to the referees for their careful and insightful comments. With their suggestions, this manuscript has been greatly improved. The referees' all comments are copied below in italics and followed by our responses. The corresponding modifications in manuscript are marked in blue color. Table 1 in the original manuscript has been renumbered as Table 2 in the revised version.

According to the suggestion of Referee #2, the newly defined stagnation index in our study is named BSI in the revision, short for Boundary-layer air Stagnation Index. Therefore, we use the name "BSI" instead of "the new index" in this response and the revision.

**Referee #1:**

Summary: This study has improved the ability of an air stagnation index to measure the atmospheric conditions of the air pollutions over China. The result is valuable and interesting and the paper is well written. I think this manuscript can meet the scope of ACP. I recommended it to be published in ACP after the following issues addressed clearly.

**Response:**

We thank the referee for the positive evaluation to this article.

**Specific Comments:**

1. As indicated in the stage of quick comment, I have pointed that this study was quite similar with a newly published paper in the journal of "Bulletin of the American Meteorological Society" (Wang et al. [kcwang@bnu.edu.cn], PM2.5 pollution in China and how it has been exacerbated by terrain and meteorological conditions, BAMS), at least, the definition of the air stagnation index. However, I have not got the available information related to the difference between them, though authors have cited this newly study. So, I still suggest the authors clarify it in the introduction.

**Response:**

Wang et al. (2017) also tried to redefine ASI to better describe the dispersion capacity of the atmosphere of China. In their definition, 10 m wind speed, boundary layer height and the occurrence of precipitation are taken into consideration. Their air stagnation threshold is determined by a fitting equation which relates to PM2.5 concentration, wind speed and boundary layer height. This equation varies with locations and changes over time. In contrast, we avoid to relate the new index directly to air pollution monitoring data. Instead, we put the thresholds on meteorological basis

only. We hope that the proposed index is more universal and robust in principle. This goal is achieved, since the BSI shows its improved ability to indicate the annual cycles of air pollution levels (Fig. 11). Furthermore, we check the day-by-day variation of the BSI to PM2.5 concentration through typical months in Beijing and it turns out that the BSI is able to track the daily variation of particulate matter (Figs. 13–15).

This response has been added in the Introduction.

References:

Wang, X., Dickinson, R. E., Su, L., Zhou, C., and Wang, K.: PM2.5 Pollution in China and How It Has Been Exacerbated by Terrain and Meteorological Conditions, Bulletin of the American Meteorological Society, doi:10.1175/bams-d-16-0301.1, 2017.

2. It has been also pointed out in the quick comment but no reflection has been reached. The hourly PM2.5 data of Beijing in January 2013 from the US Embassy is used here. However, in general, this sort of data cannot be used in the open published paper because this monitoring is not a regularly site-observation.

**Response:**

Hourly concentration of PM2.5 data during January 2013 in Beijing are not yet accessible from Ministry of Environmental Protection (MEP) of China (http://datacenter.mep.gov.cn/). So we analyzed the data from US Embassy instead. According to Liang et al. (2016), PM2.5 data from the US post and MEP stations in Beijing (i.e., Dongsi, Nongzhanguan, Dongsihuan, the nearest stations to the US Embassy station) were highly consistent. Moreover, data from US Embassy have already been used in many open published papers such as Cai et al. (2017) and Lv et al., (2017). Therefore, we considered PM2.5 data from US Embassy be reliable. Besides, we have added more case studies of Beijing during 2015–2017 in this revision, using 24-h averaged PM2.5 concentrations from MEP of China (Figs. 14 and 15).

References:

- Cai, W. J., Li, K., Liao, H., Wang, H. J., and Wu, L. X.: Weather conditions conducive to Beijing severe haze more frequent under climate change, Nature Climate Change, 7, 257-+, doi:10.1038/nclimate3249, 2017.
- Liang, X., Li, S., Zhang, S. Y., Huang, H., and Chen, S. X.: PM2.5 data reliability, consistency, and air quality assessment in five Chinese cities, Journal of Geophysical Research-Atmospheres, 121, 10220-10236, doi:10.1002/2016jd024877, 2016.
- Lv, B. L., Cai, J., Xu, B., and Bai, Y. Q.: Understanding the Rising Phase of the PM2.5 Concentration Evolution in Large China Cities, Scientific Reports, 7, doi:10.1038/srep46456, 2017.

3. Some newly works in this aspect should be reviewed. For example: (1) Cai WJ et al., 2017: Weather conditions conducive to Beijing severe haze more frequent under climate change. Nature Climate Change, doi:10.1038/NCLIMATE3249. (2) Yin ZC et al., 2017: Understanding severe winter haze events in the North China Plain in 2014: roles of climate anomalies. Atmos. Chem. Phys., 17, 1641-1651. (3) Han ZY et al., 2017:

Projected changes in haze pollution potential in China: an ensemble of regional climate model simulations. Atmos. Chem. Phys., 17, 10109-10123. (4) Wang HJ et al., 2016: Understanding the recent trend of haze pollution in eastern China: roles of climate change. Atmos. Chem. Phys., 16, 4205-4211.

**Response:**

Accepted. These studies have been reviewed in the Introduction.

Studies have shown that haze pollutions in China occurs more frequently over the past decades, especially those in the eastern region (Wang and Chen, 2016; Pei et al., 2018). Apart from increased emissions of pollutants, climate change also plays an important role in air pollution intensification. Less cyclone activities and weakening East Asian winter monsoons (Pei et al., 2018; Yin et al., 2017) resulted from the decline of Arctic sea ice extent (Deser et al., 2010; Wang and Chen, 2016) and expanded tropical belt (Seidel et al., 2008) are believed to bring about more stable atmosphere in eastern China and cause more haze days (Wang et al., 2015). Hence, the analysis of meteorological background related to air pollution is of great importance.

Researchers have developed many indexes comprised of different meteorological variables to describe the ability of transport and dispersion of the atmosphere. For example, Cai et al. (2017) constructed haze weather index with 500 hPa and 850 hPa wind speeds and temperature anomalies between 850 hPa and 250 hPa. Han et al. (2017) proposed air environment carrying capacity index including air pollutants concentration, precipitation and ventilation coefficient. Among these indexes, air stagnation index (ASI), consisting of upper- and lower-air winds and precipitation, is in use till today. The US National Climatic Data Center (NCDC) monitors monthly air stagnation days for the United States since 1973 to indicate the temporal buildup of ozone in the lower atmosphere (http://www.ncdc.noaa.gov/societal-impacts/air-stagnation/).

**References:**

- Cai, W., Li, K., Liao, H., Wang, H., and Wu, L.: Weather conditions conducive to Beijing severe haze more frequent under climate change, Nature Climate Change, 7, 257-262, 2017.
- Deser, C., Tomas, R., Alexander, M., and Lawrence, D.: The seasonal atmospheric response to projected Arctic sea ice loss in the late 21st century, J. Climate, 23, 333–351, 2010.
- Han, Z., Zhou, B., Xu, Y., Wu, J., and Shi, Y.: Projected changes in haze pollution potential in China: an ensemble of regional climate model simulations, Atmospheric Chemistry and Physics, 17, 10109-10123, 2017.
- Pei, L., Yan, Z., Sun, Z., Miao, S., and Yao, Y.: Increasing persistent haze in Beijing: potential impacts of weakening East Asian winter monsoons associated with northwestern Pacific sea surface temperature trends, Atmospheric Chemistry and Physics, 18, 3173-3183, 2018.
- Seidel, D. J., Fu, Q., Randel, W. J., and Reichler, T. J.: Widening of the tropical belt in a changing climate, Nat. Geosci., 1, 21–24, https://doi.org/10.1038/ngeo.2007.38, 2008.
- Wang, H.-J., and Chen, H.-P.: Understanding the recent trend of haze pollution in eastern China: roles of climate change, Atmospheric Chemistry and Physics, 16, 4205-4211, 2016.
- Wang, H. J., Chen, H. P., and Liu, J. P.: Arctic sea ice decline intensified haze pollution in eastern China, Atmos. Ocean. Sci. Lett., 8, 1–9, 2015.

Yin, Z., Wang, H., and Chen, H.: Understanding severe winter haze events in the North China Plain in 2014: roles of climate anomalies, Atmospheric Chemistry and Physics, 17, 1641, 2017.

4. Some methodology explanations should be added. For example, "temperature profiles from radiosonde are linearly interpolated to 1-m vertical intervals", "Wind profile from 1200 UTC (i.e., 2000 BJT) sounding data is also interpolated to 1-m vertical grids, ": : : : : how to complete it? Different results may be obtained when different methods used.

**Response:**

The data of temperature and wind speed from radiosonde were interpolated linearly to height levels with 1-m vertical resolution. For the wind speeds under 100 m, logarithmic interpolation might be more appropriate. Linear interpolation may result in an underestimation of wind speeds and thus an underestimation of ventilation. But this can be negligible compared to the large value of daily maximum ventilation.

The sentence has been rewritten in this revision.

5. In this study, just 66 stations are used across China which is resampled into 2\*2 grids. Obviously, there are many grids that even have no stations, resulting in misleading to readers for the information over these grids. Of course, some descriptions in this MS are not precise. For example, there is just one station in Tibet Plateau and the information of the spatial patterns for the ventilation, CAPE etc. are just the interpolation results, cannot represent the actual distribution. So, the authors say that the ventilation (CAPE etc.) is largest over Tibet Plateau may be not correct.

**Response:**

Available observation data are indeed very limited over Qinghai–Tibet Plateau. Our results over this area are not very precise since the observation stations are sparse. Therefore, some of our descriptions need to be modified to be more modest. But in general, our results are reasonable, as explained below.

First, the interaction between the plateau and the atmosphere has long been recognized. High terrain elevation corresponds to thinner air and lower value of aerosol optical thickness, less precipitable water vapor amount, and stronger solar radiation (Chen et al., 2014; Zhu et al., 2010), which enhance thermal convections and lead to very high mixing layer over the plateau. The results agree with the recent study of Liu et al. (2015). On the other hand, the plateau locates right in the westerlies, and it can even break the jet stream into two branches surrounding the northern and southern edges of the plateau. Therefore, as Schiemann et al. (2009) revealed, the transport wind is strong (more than 6 m/s) over plateaus because of the combined influence of westerly jet and elevated topography. As a result, ventilation is largest over Tibet Plateau as a combined effect of the large mixing layer height and strong transport wind.

Second, there is also large CAPE in the atmosphere over the southern Qinghai– Tibet Plateau, when the summer monsoon of south Asia is established, with warm and moist air mass originating from the Bay of Bengal being blown to Tibet plateau (Romatschke et at., 2010). We have rewritten related sentences in this revision.

References:

- Chen, J.-L., Xiao, B.-B., Chen, C.-D., Wen, Z.-F., Jiang, Y., Lv, M.-Q., Wu, S.-J., and Li, G.-S.: Estimation of monthly-mean global solar radiation using MODIS atmospheric product over China, Journal of Atmospheric and Solar-Terrestrial Physics, 110, 63-80, doi:10.1016/j.jastp.2014.01.017, 2014.
- Liu, J., Huang, J., Chen, B., Zhou, T., Yan, H., Jin, H., Huang, Z., and Zhang, B.: Comparisons of PBL heights derived from CALIPSO and ECMWF reanalysis data over China, Journal of Quantitative Spectroscopy and Radiative Transfer, 153, 102-112, 2015.
- Romatschke, U., Medina, S., and Houze, R. A., Jr.: Regional, Seasonal, and Diurnal Variations of Extreme Convection in the South Asian Region, Journal of Climate, 23, 419-439, doi:10.1175/2009jcli3140.1, 2010.
- Schiemann, R., Lüthi, D., and Schär, C.: Seasonality and interannual variability of the westerly jet in the Tibetan Plateau region, Journal of Climate, 22, 2940-2957, 2009.
- Zhu, X., He, H., Liu, M., Yu, G., Sun, X., and Gao, Y.: Spatio-temporal variation of photosynthetically active radiation in China in recent 50 years, Journal of Geographical Sciences, 20, 803-817, 2010.

6. "Another discrepancy is the high ASI in October and November in Urumqi, corresponding to relatively lower API values." I suggest the authors to check the variation of each component of ASI. May be it resulted by one of the components. **Response:**

According to the reviewer's suggestion, we check each component of the BSI.

As shown in Figs. 4, 6 and 8 in the manuscript, ventilation condition in Urumgi is good from April to September with average values larger than  $10000 \text{ m}^2/\text{s}$ , and becomes much worse from October to March in the next year with average values less than 5000  $m^2/s$ . We counted days with poor ventilation conditions (i.e. daily maximum ventilation is less than 6000 m2/s) in every month during 1985 to 2014 (Fig. R1), and found that poor ventilation conditions are rarely happen during April to September but occur frequently during October to March. The annual cycle of poor ventilation days matches well with that of BSI. The other two components CAPE CIN and precipitation days keep a rather constant value yearly around (Figs. 6 and 8). So, we confirm that the poor ventilation is the dominant contributor for high BSI in October and November in Urumqi.

But we are still unable to explain the different trend between BSI and API during October to December in Urumqi: the ventilation condition is almost equally poor in these three months; whereas, the API grows significantly from October to December.

Figure R1. Annual cycle of poor ventilation days (i.e. days with ventilation less than  $6000 \text{ m}^2\text{/s}$ ).

7. "In order to exclude the influences of emissions as much as possible, the investigation only covers data of winter half-year (i.e., October–March) when domestic heating requires more energy consumption." This sentence confused me.

**Response:**

This sentence has been rewritten as follows:

"In order to exclude the impact of seasonal variation in source strengths, the investigation only covers data of winter half-year (i.e., October–March) when domestic heating in north China leads to more energy consumption and more serious air pollution pressure."

8. In this study, the newly developed ASI is compared with the original one and the results indicating a better performance for newly index to capture the air stagnation days. The correlation coefficients should be shown in the text that can increase the readability.

**Response:**

Accepted. The correlation coefficients have been added in Fig. 12.

9. From the comparison between the newly and original ASI, we can find that there are generally peak stagnation days in summer from original ASI but winter from newly one. Why?

**Response:**

In general, stagnation days under ASI metric is more in summer and less in winter. According to Huang et al. (2017), the behavior of upper-air wind speeds is the main driver of the ASI distribution. A weaker latitudinal pressure gradient at the upper layer of the atmosphere results in more air stagnation occurrences in summer. In contrast, the BSI is observed to be maximum in winter and minimum in summer. This behavior results from the cumulative responses of its components. We use standardized partial regression coefficients from multiple linear regression analysis to determine which variable is more important (Table R1, please see the answers to Questions 2 from Referee #2). It is shown that during the wintertime of Harbin, Urumqi and Beijing, ventilation is the main driver of the BSI; precipitation contributes less; and cumulus convection barely plays a part. In summer, however, ventilation and cumulus convection both contribute a lot. For Harbin, they play an almost equal role, with regression coefficients of 0.57 and 0.48 respectively. For Beijing, the contribution of CAPE out weights that of ventilation more than twice. For Urumqi which is in the semiarid climate region, the ventilation is still the main driver of the behavior of the BSI ( $\beta$  =0.83), while precipitation and CAPE contribute equally ( $\beta$ =0.36 and 0.4 respectively). The characteristics in Chongqing is different from the other three stations. With more convective energy stored in the atmosphere (Fig. 8), the seasonal variation of the BSI relies more on CAPE: it is the second contributor during winter and spring ( $\beta$ =0.36 and 0.38 respectively) and becomes the main contributor in summer and autumn ( $\beta$ =0.75 and 0.68 respectively). Noticeably in summer, the BSI is dominated by cumulus convection, while the ventilation barely plays a role.

The intention we add CAPE/CIN into the criteria is to make up for the deficiency of daily maximum mixing layer depth (MMD) derived from parcel method failing to describe the convection between the cloud base and cloud top. Therefore, it is expected that CAPE/CIN plays a more important role in summer.

This explanation has been added in Section 4.

References:

Huang, Q., Cai, X., Song, Y., and Zhu, T.: Air stagnation in China (1985–2014): climatological mean features and trends, Atmospheric Chemistry and Physics, 17, 7793-7805, 2017.

**10. From Figure 12, we can see that the numbers of stagnation days are generally much larger from the newly ASI than the original one. Why?**

**Response:**

Figure 12 only displayed stagnant conditions of winter half-year (i.e., October–March). The BSI is larger in winter and smaller in summer, while the ASI is larger in summer (Fig. 10). Therefore, when we compare the original and new stagnation index in the winter half-year, the numbers of stagnation days are generally much larger for the BSI. The explanation has been added in the manuscript.

11. Some figure captions are not clear. Please check it. For example, what's the mean of the whisker in Figure 11.

**Response:**

Monthly mean values of API are given as horizontal bars in the middle, 25% and 75% percentiles are shown as boxes' lower and upper boundaries, and 10% and 90% percentiles as lower and upper whiskers. This explanation has been added in the figure caption.

**Referee #2:**

Synopsis: The authors design an atmospheric stagnation index for use in China, contrast it with the U.S. NCDC ASI, and suggest that theirs better captures the meteorological conditions conducive to poor AQ. Recommendation: This is an

interesting and needed study. The disconnect between AQ observations and the NCDC ASI over China has been shown in many previous publication. This, of course, has motivated many to design a better approach. In this attempt the authors propose meteorological factors that integrate scavenging, dispersal, and ventilation effects. I find the meteorological basis of the index design to be sound. However, I think its application and assessment require greater rigor, while the manuscript itself could be better organized and written.

**Response:**

The referee's positive evaluation is very much appreciated.

**My editorial and scientific suggestions follow:**

1. From a communication standpoint, I would advise the authors to clearly establish a new metric, i.e., do not call your index the "new ASI." The author's should look at this as an opportunity: : : an opportunity that will assist them in improving the clarity of the manuscript.

**Response:**

We thank the referee for this advice. In the revised manuscript, the new ASI is called BSI, short for Boundary-layer air Stagnation Index.

2. The creation of this index is ripe for multiple linear regression analysis. The authors suggest the ASI is overly reliant on 500mb winds. It seems likely that this new index may have similar issues. For example, if the CAPE/CIN is often zero, perhaps it provides little value to the calculation. With multiple linear regression analysis, one could quantitatively learn the value each new component adds to the overall result. I would advise this analysis for both monthly and daily data, where available.

**Response:**

The intention we add CAPE/CIN into the criteria is to make up for the deficiency of daily maximum mixing layer depth (MMD) derived from parcel method failing to describe the convection between the cloud base and cloud top. Therefore, it is expected that CAPE/CIN plays a more important role in summer.

We use standardized partial regression coefficients from multiple linear regression analysis to determine which variable is more important to BSI results (Table R1). It is shown that during the wintertime of Harbin, Urumqi and Beijing, ventilation is the main driver of BSI; precipitation contributes less; and cumulus convection barely plays a part in BSI results. In summer, however, ventilation and cumulus convection both contribute a lot. For Harbin, they play an almost equal part, with regression coefficients of 0.57 and 0.48 respectively. For Beijing, the contribution of CAPE out weights that of ventilation more than twice. For Urumqi which is in the semi-arid climate region, the ventilation is still the main driver of BSI behavior ( $\beta$ =0.83), while precipitation and CAPE contribute equally ( $\beta$ =0.36 and 0.4 respectively). The characteristics in Chongqing is different from the other three stations. With more convective energy stored in the atmosphere (Fig. 8), the seasonal variation of BSI relies more on CAPE: it is the second contributor during winter and spring ( $\beta$ =0.36 and 0.38 respectively) and becomes the main contributor in summer and autumn ( $\beta$ =0.75 and 0.68 respectively). Noticeably in summer, the BSI is dominated by cumulus convection, while the ventilation barely plays a role.

This explanation has been added in Section 4 and Table R1 has been added in the revised manuscript as Table 1.

|           |       |                                                          | CALL.       |          |                |  |
|-----------|-------|----------------------------------------------------------|-------------|----------|----------------|--|
| Station   | Month | Standardized Partial Regression Coefficients ( $\beta$ ) |             |          | $\mathbf{P}^2$ |  |
|           |       | Precipitation                                            | Ventilation | CAPE_CIN | ĸ              |  |
| Harbin    | Jan.  | 0.22                                                     | 1.00        | -0.02    | 0.99           |  |
|           | Apr.  | 0.58                                                     | 0.84        | -0.13    | 0.74           |  |
|           | Jul.  | 0.30                                                     | 0.57        | 0.48     | 0.70           |  |
|           | Oct.  | 0.37                                                     | 0.98        | 0.01     | 0.95           |  |
| Urumqi    | Jan.  | 0.17                                                     | 1.00        | -0.03    | 0.99           |  |
|           | Apr.  | 0.23                                                     | 1.04        | 0.10     | 0.89           |  |
|           | Jul.  | 0.36                                                     | 0.83        | 0.40     | 0.77           |  |
|           | Oct.  | 0.33                                                     | 0.91        | 0.12     | 0.95           |  |
| Beijing   | Jan.  | 0.30                                                     | 1.06        | 0.04     | 0.98           |  |
|           | Apr.  | 0.45                                                     | 0.85        | 0.05     | 0.82           |  |
|           | Jul.  | 0.07                                                     | 0.33        | 0.70     | 0.47           |  |
|           | Oct.  | 0.26                                                     | 0.85        | 0.19     | 0.80           |  |
| Chongqing | Jan.  | 0.21                                                     | 0.63        | 0.36     | 0.98           |  |
|           | Apr.  | 0.32                                                     | 0.56        | 0.38     | 0.89           |  |
|           | Jul.  | 0.21                                                     | -0.02       | 0.75     | 0.67           |  |
|           | Oct.  | 0.35                                                     | 0.25        | 0.68     | 0.84           |  |

Table R1. Dependence of BSI on its components, i.e. daily precipitation, ventilation and CAPE.a

a The relationships between monthly stagnation days and its corresponding components were derived by using multiple linear regression analysis. "Precipitation" stands for number of days with no precipitation; "Ventilation" for number of days when daily maximum ventilation is less than 6000 m2/s; "CAPE\_CIN" for number of days when the value of CAPE is less than the absolute value of CIN.

3. From a fine temporal resolution perspective, I appreciate the focused analysis on Beijing, however I wonder why this portion of the analysis wasn't extended into other seasons and time periods. Does the "new ASI" perform as well during poor AQ events in the summer/spring/fall seasons? If daily Embassy data is sufficient to analyze in January of 2013, surely you could test your metric on other time periods.

**Response:**

Accepted. We have tested the performance of BSI on other time periods and results are shown below.